# Repressive H3K9me2 protects lifespan against the transgenerational burden of COMPASS activity in *C. elegans*

Teresa Wei-sy Lee, Heidi Shira David, Amanda Kathryn Engstrom, Brandon Scott Carpenter, David John Katz*

Department of Cell Biology, Emory University School of Medicine, Atlanta, United States

**Abstract** In *Caenorhabditis elegans*, mutations in WDR-5 and other components of the COMPASS H3K4 methyltransferase complex extend lifespan and enable its inheritance. Here, we show that *wdr-5* mutant longevity is itself a transgenerational trait that corresponds with a global enrichment of the heterochromatin factor H3K9me2 over twenty generations. In addition, we find that the transgenerational aspects of *wdr-5* mutant longevity require the H3K9me2 methyltransferase MET-2, and can be recapitulated by removal of the putative H3K9me2 demethylase JHDM-1. Finally, we show that the transgenerational acquisition of longevity in *jhdm-1* mutants is associated with accumulating genomic H3K9me2 that is inherited by their long-lived wild-type descendants at a subset of loci. These results suggest that heterochromatin facilitates the transgenerational establishment and inheritance of a complex trait. Based on these results, we propose that transcription-coupled H3K4me via COMPASS limits lifespan by encroaching upon domains of heterochromatin in the genome.

## Introduction

Lifespan is governed by complex interactions between genetics and the environment. Despite its complexity as a trait, lifespan is limited by the germline in a wide range of metazoans. For example, genetic or physical ablation of the germline extends lifespan in *C. elegans* (*Arantes-Oliveira, 2002*; *Flatt et al., 2008*; *Hsin and Kenyon, 1999*; *Kenyon, 2010*). These observations led to the disposable soma theory of aging, which posits that resources are shifted to the germline to promote progeny fitness at the expense of maintaining the parental soma (*Kenyon, 2010*; *Kirkwood and Holliday, 1979*). Some have theorized that one cost of a germline is transcription during gametogenesis, but the molecular details of how germline transcription affects lifespan remain elusive (*Ghazi et al., 2009*; *Greer et al., 2010*).

Regions of active transcription are marked by post-translational histone modifications, like the methylation of histone 3 at lysine 4 (H3K4me) (*Shilatifard, 2008*). H3K4me is deposited by the MLL/COMPASS complex, which travels with elongating RNA Polymerase II during transcription (*Wood et al., 2007*). In the nematode *C. elegans*, animals with reductions in COMPASS complex subunits (*wdr-5, ash-2,* and the methyltransferase *set-2*) live longer than wild-type individuals (*Greer et al., 2010*). The lifespan extension of COMPASS mutants requires the presence of an actively proliferating germline, suggesting that COMPASS acts specifically in the germline to limit lifespan (*Greer et al., 2010*). Subsequent work showed that reducing COMPASS solely in the germline activates a fatty acid desaturation pathway in the intestine that increases lifespan, in part by down-regulating an S6 kinase normally expressed in the germline (*Han et al., 2017*). Activation of this pathway causes the accumulation of mono-unsaturated fatty acids, and this increase is sufficient to extend lifespan (*Han et al., 2017*).

*For correspondence:
djkatz@emory.edu

Competing interests: The authors declare that no competing interests exist.

Greer and colleagues also found that longevity could be heritable. The lifespan extension of COMPASS mutants was inherited by their wild-type descendants for four generations, before reverting to wild-type lifespan (*Greer et al., 2011*). The transgenerational inheritance of longevity suggests that it is an epigenetic trait, but the mechanism of inheritance remains unknown. In particular, it is difficult to reconcile how reduced H3K4me in COMPASS mutants can block COMPASS from restoring a normal chromatin environment in genetically wild-type descendants of COMPASS mutants. Thus, the longevity of COMPASS mutants may be due to another heritable factor.

We have previously described an epigenetic reprogramming mechanism in which two chromatin modifiers cooperate to erase traces of prior transcription at fertilization. The demethylase SPR-5 (LSD1/KDM1A) removes active H3K4me1/2, while the methyltransferase MET-2 (SETDB1) methylates histone 3 at lysine 9 (H3K9me) (*Katz et al., 2009*; *Kerr et al., 2014*). This reprogramming mechanism responds to reduced H3K4me in COMPASS mutants by adding more H3K9me2 (*Kerr et al., 2014*). We therefore wondered whether the longevity of COMPASS mutants could be associated with H3K9me2 deposited in response to reductions in transcription-coupled H3K4me.

H3K9me2 is classically considered a repressive modification and often associated with heterochromatin. In *C. elegans*, H3K9me2 and H3K9me3 each have distinct roles in the genome, with H3K9me2, rather than H3K9me3, most closely associated with canonical heterochromatin factors like HP1 (*Garrigues et al., 2015*; *Liu et al., 2011*; *McMurchy et al., 2017*). One theory of aging, the heterochromatin loss model, is supported by several lines of evidence suggesting that losing repressive chromatin is detrimental for lifespan (*Tsurumi and Li, 2012*; *Villeponteau, 1997*). In humans, two premature aging diseases are caused by mutations in lamins that reduce heterochromatin and disrupt its nuclear localization (*Shumaker et al., 2006*). In *C. elegans*, *Drosophila*, and mammals, heterochromatin decreases as individuals grow older (*Wood and Helfand, 2013*; *Wood et al., 2010*; *Haithcock et al., 2005*). Additionally, across eukaryotes, mutations that increase repressive chromatin extend lifespan (*Jin et al., 2011*; *Kennedy et al., 1995*; *Larson et al., 2012*; *Maures et al., 2011*). Therefore, we investigated the interaction between repressive H3K9me2 and *wdr-5* mutant lifespan over generational time.

## Results

### Transgenerational longevity in *wdr-5* mutants

It has previously been reported that animals mutant for genes encoding components of the COMPASS complex have a lifespan extension of up to 28% (*Greer et al., 2010*).This lifespan extension is inherited by wild-type descendants of these mutants, before reverting back to wild-type levels in the fifth generation (*Greer et al., 2011*). To investigate the nature of this inheritance, we first attempted to recapitulate the original observation using fertile, homozygous *wdr-5 (ok1417)* mutants. Since the transgenerational effects of a *wdr-5* mutation lasts for four generations, we crossed *wdr-5* mutants to wild type to generate heterozygous *wdr-5/+* progeny and maintained populations as *wdr-5/+* heterozygotes for five generations. We then used homozygous *wdr-5* mutant progeny descended from F5 *wdr-5/+* heterozygotes as our P0 founding population, comparing them to P0 wild-type animals descended from survivors recovered from a thaw. In contrast to prior observations, P0 *wdr-5* mutants were never long-lived compared to their wild-type counterparts (as observed in eight biological replicates) (*Figure 1*, *Figure 1—figure supplement 1*). Furthermore, in contrast to what was originally reported (*Greer et al., 2010*), treatment of wild-type animals with *wdr-5* RNA interference (RNAi) did not cause increased lifespan within a single generation (*Figure 1—figure supplement 3*).

The initial observations of *wdr-5* mutant longevity did not report how many generations lacking WDR-5 activity were required to reach the 28% lifespan extension in *wdr-5* mutants, so we considered the possibility that lifespan may gradually change over a number of generations in COMPASS mutants. Starting with P0 as described above, we followed a single population of *wdr-5* mutants and assessed lifespan periodically. For the first six generations (which are hereafter referred to as early-gen populations), *wdr-5* mutants had slightly shorter lifespans than wild-type populations of the same generation, though the decrease was not statistically significant (*Figure 1*, *Figure 1—figure supplement 1*). In generations F8 to F14 (hereafter referred to as mid-gen populations), *wdr-5* mutants had lifespans that were the same as, or slightly longer than, wild-type populations (*Figure 1*), though the change was only statistically significant in some generations (*Figure 1—figure*

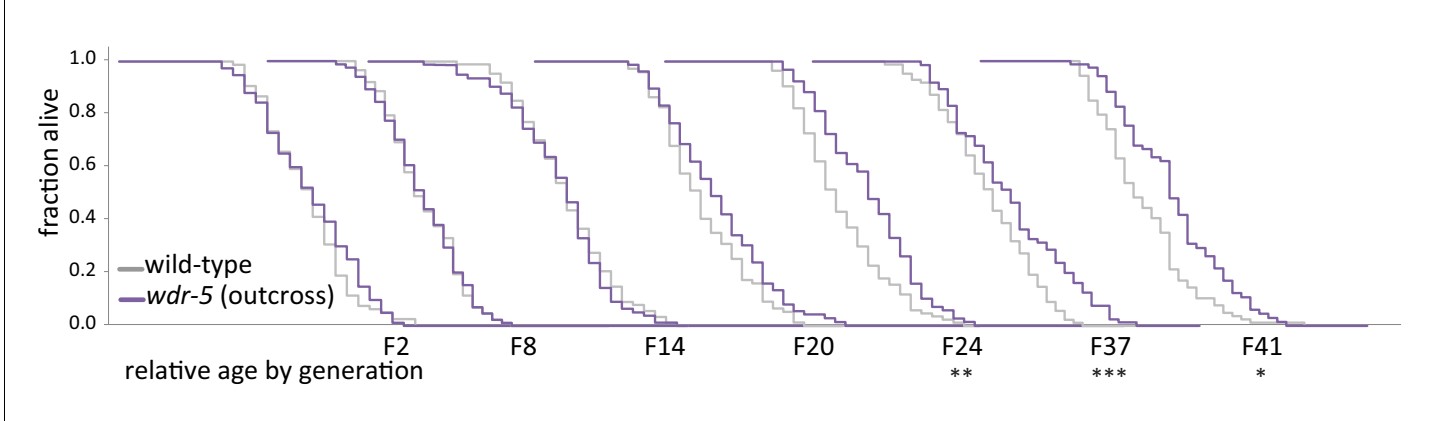

**Figure 1.** Longevity takes many generations to manifest in *wdr-5* mutants. Analysis of relative lifespan between wild-type (gray) and *wdr-5* mutants (purple) across generational time. P0 wild type was descended from animals recovered from a thaw. P0 *wdr-5* mutants were the first homozygous mutants after maintenance as heterozygotes for five generations. For each generation, the x-axis is 40 days. *p<0.05, **p<0.01, ***p<0.001 compared to wild-type from the same generation with log-rank test. Median lifespan and statistics are presented in *supplementary file 1*. Additional replicates shown in *Figure 1—figure supplement 1* and *supplementary file 3*.

The online version of this article includes the following figure supplement(s) for figure 1:

**Figure supplement 1.** Longevity in *wdr-5* mutants reproducibly takes many generations to manifest.

**Figure supplement 2.** Longevity is not caused by decreased fecundity or by background mutations.

**Figure supplement 3.** Depletion by RNAi does not increase lifespan within twelve generations.

---

*supplement 1*). By generation F20 and in subsequent generations (hereafter referred to as late-gen populations), *wdr-5* mutant populations were consistently and statistically longer-lived than wild-type populations of the same generation (*Figure 1*, *Figure 1—figure supplement 1*). Additionally, populations of wild-type animals continuously treated with *wdr-5* RNAi experienced a similarly gradual increase in lifespan, with a modest increase seen by generation F12 (*Figure 1—figure supplement 3*, p=0.039, log-rank test). Overall, late-gen *wdr-5* mutants had median lifespan extensions ranging from 5–40% (with an average of 17% across multiple replicates) which recapitulated the 16–28% extension originally reported (*supplementary files 1* and *3*) (*Greer et al., 2010*; *Greer et al., 2011*). Thus, we conclude that *wdr-5* mutants had longer lifespans, but only after many generations of lacking WDR-5 activity.

In many taxa, fecundity is inversely correlated with lifespan (*Kenyon, 2010*). To determine whether the appearance of longevity in *wdr-5* mutant populations correlates with a decrease in fecundity, we measured progeny number across generational time. It has previously been shown that *wdr-5* mutants have decreased broods and increased embryonic lethality (*Li and Kelly, 2011*; *Simonet et al., 2007*). Consistent with prior observations, early-gen *wdr-5* mutants had 27–47% fewer progeny than wild-type populations (*supplementary file 2*). However, we did not observe a correlation between the onset of longevity and the decrease in progeny number, as long-lived late-gen *wdr-5* mutants did not have significantly fewer progeny than early- or mid-gen populations (*Figure 1—figure supplement 2A and B*, p=0.11 and p=0.13, respectively, unpaired t-test). This result suggests that lifespan extension in *wdr-5* mutants was not a direct consequence of reduced fecundity, despite the fact that it requires a proliferative germline (*Greer et al., 2010*).

The appearance of longevity in *wdr-5* mutants could be caused by the acquisition of background mutations in lifespan-determining genes, but several observations make this scenario unlikely. First, we have repeated the transgenerational analysis of lifespan in these populations seven additional times. In each replicate, lifespan increased gradually between early-, mid-, and late-gen populations, with late-gen *wdr-5* mutants consistently living longer than late-gen wild-type populations (*Figure 1—figure supplement 1*). Additionally, when long-lived late-gen *wdr-5* mutants were subjected to either starvation (*Figure 1—figure supplement 2C*) or freezing (*Figure 2*), lifespan reverted back to wild-type levels. We have also outcrossed long-lived late-gen *wdr-5* mutants to wild-type populations, maintained these populations as heterozygotes for five generations, and re-selected new homozygous P0 *wdr-5* mutant populations. These P0 *wdr-5* mutants reverted back to being short-

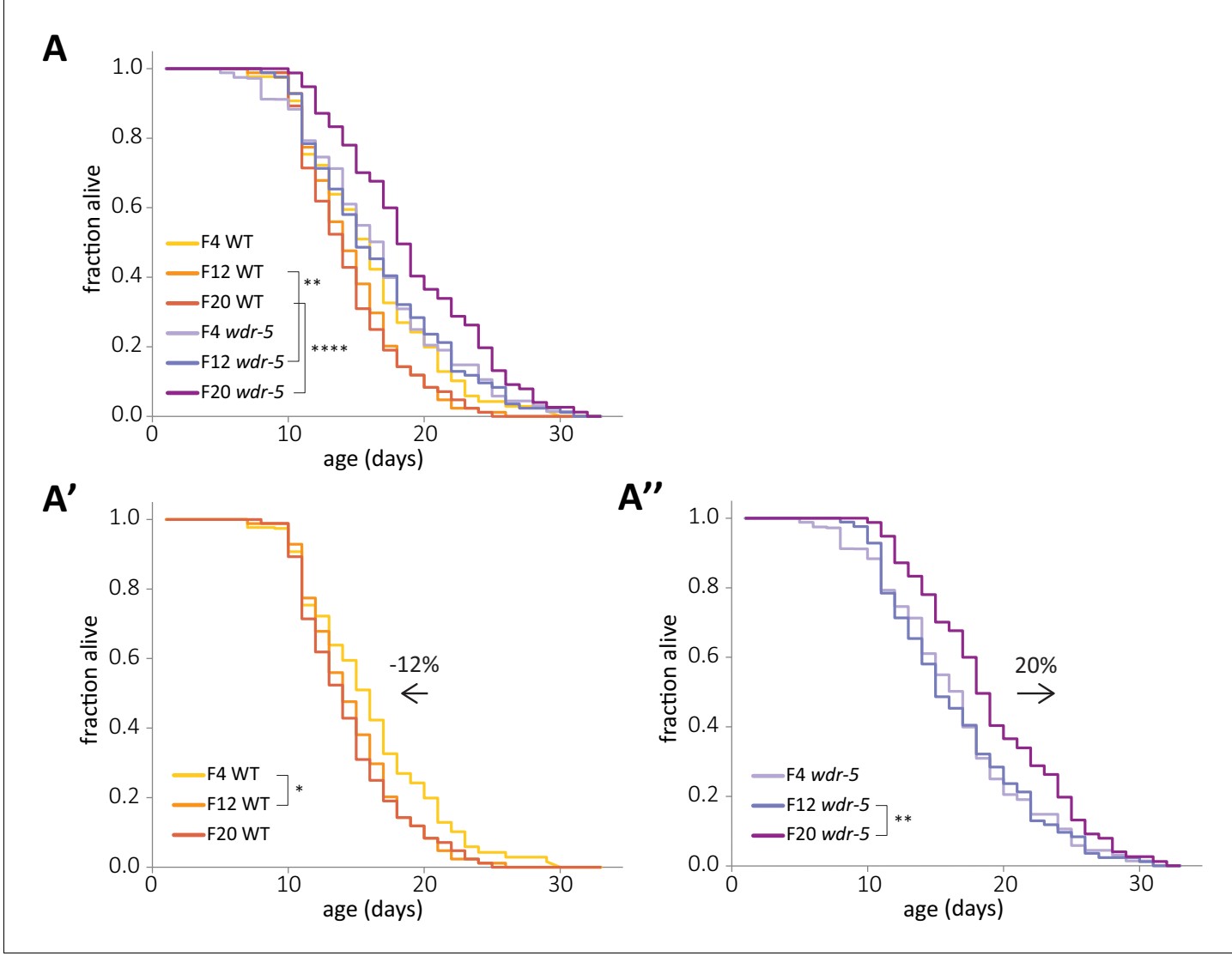

**Figure 2.** Concurrent and opposite changes in lifespan account for the full *wdr-5* mutant lifespan extension. (**A**) Lifespan of early-, mid-, and late-gen wild-type (yellow, tangerine, and burnt orange, respectively) and *wdr-5* mutant populations (lavender, purple, and plum, respectively) descended from animals recovered from a thaw. Data are also shown separated into wild-type (**A'**) and *wdr-5* mutant populations (**A''**). Percentage difference in median lifespan between early- and late-gen is indicated above arrow. p<0.05, **p<0.01, and ****p<0.0001 using log-rank test. Median lifespan and statistics are presented in *supplementary file 1*. Additional replicates are included in *supplementary file 3*.

The online version of this article includes the following figure supplement(s) for figure 2:

**Figure supplement 1.** Lifespan does not differ between populations descended from the same outcross.

lived, even as the original late-gen *wdr-5* mutant populations remained long-lived (*Figure 1—figure supplement 2D*).

## Relative changes in lifespan across generations

The determination of longevity in a population is always relative to a control. In our transgenerational lifespan experiments, *wdr-5* mutant populations were compared to wild-type populations of the same generation. Therefore, the appearance of longevity could be caused by a relative extension in *wdr-5* mutant lifespan, a relative decrease in wild-type lifespan, or changes in both populations. To distinguish between these possibilities, we directly compared the lifespan of early-, mid-, and late-gen *wdr-5* mutant and wild-type populations in the same lifespan assay (*Figure 2A*). Surprisingly, wild-type lifespan decreased over generational time by 12%, primarily between early- and

mid-gen populations (*Figure 2A'*, p=0.02, log-rank test), despite the fact that progeny number did not change in these populations (*supplementary file 2*). As mentioned previously, we assayed wild-type lifespan in descendants of animals recovered from a thaw, which we consider the P0 population. The process of freezing a strain involves an L1 larval diapause induced by starvation (*Baugh, 2013*). The generational change in wild-type lifespan after recovery from a thaw suggested that some aspect of starvation may increase lifespan in descendants, as has been previously shown in populations recovered from L1 diapause or from dauer diapause (*Rechavi et al., 2014*; *Webster et al., 2018*). To investigate the effect of starvation on wild-type populations, we starved a late-gen population that had already experienced a decrease in lifespan. Wild-type descendants of starved animals lived longer than their non-starved cousins (*Figure 1—figure supplement 2C*, p=0.001, log-rank test). To further confirm that the increase in wild-type lifespan in populations recovered from a freeze was caused by the freeze itself, we compared *wdr-5* mutants and wild-type populations that were both descended from the same outcrossed population that had been maintained as *wdr-5/+* heterozygotes for five generations. If the initial increase in wild-type lifespan was caused by the freeze-thaw process, then wild-type populations derived from the outcross, which did not undergo a freeze-thaw, should not have this initial increase in lifespan. Consistent with this possibility, we found that after the outcross, both early-gen wild-type animals and *wdr-5* mutants had similar lifespans (*Figure 2—figure supplement 1*, p=0.32, log-rank test). In addition, early-gen wild-type animals from the outcross had a shorter lifespan than early-gen wild-type animals from a thaw (*Figure 2—figure supplement 1*, p=0.02, log-rank test). These results confirmed that wild-type lifespan was only extended after undergoing the freeze-thaw process.

In contrast to what occurred in wild-type animals, lifespan in *wdr-5* mutants increased over generational time by 20%, primarily between mid- and late-gen populations (*Figure 2A''*, p=0.009, log-rank test). The decrease in wild-type lifespan was large enough that even early-gen *wdr-5* mutants had significantly longer lifespans when compared to those of late-gen wild-type populations (*Figure 2A*, p=0.019, log-rank test). However, changes in wild-type lifespan were not the sole driver of longevity in late-gen *wdr-5* mutants, since late-gen *wdr-5* mutants lived significantly longer than even early-gen wild-type populations (p=0.0009, log-rank test). Overall, concurrent and opposite lifespan changes in both wild type and *wdr-5* mutants accounted for the full increase in lifespan in late-gen *wdr-5* mutant populations.

## Correlation of longevity with repressive H3K9me2

Our previous work demonstrates that levels of H3K9me2 are enriched at certain loci in *wdr-5* mutants compared to wild type (*Kerr et al., 2014*). Based on this finding, we asked whether an increase in H3K9me2 could account for the extended lifespan of *wdr-5* mutants. To address this possibility, we first compared global levels of H3K9me2 by immunoblot and chromatin immunoprecipitation followed by next-generation sequencing (ChIP-seq). By immunoblot of mixed-stage populations, long-lived late-gen *wdr-5* mutants had a global increase of H3K9me2 compared to late-gen wild-type populations (*Figure 3A*).

We next examined the genomic enrichment of H3K9me2 using ChIP-seq across generations in *wdr-5* mutant and wild-type populations. At H3K9me2 peaks, early-gen *wdr-5* mutants had slightly less enrichment than their wild-type counterparts (*Figure 3B*). In wild type, H3K9me2 enrichment decreased (*Figure 3B'*) at the same time that lifespan decreased (*Figure 2A'*) between early- and mid-gen populations. Between mid- and late-gen wild-type populations, H3K9me2 enrichment subsequently decreased further as wild-type lifespan remained at steady-state (*Figure 3B'*). *wdr-5* mutants experienced a similar H3K9me2 decrease between early- and mid-gen populations, as lifespan remained unchanged. However, H3K9me2 enrichment then increased between mid- and late-gen populations (*Figure 3B''*), concomitant with the increase in lifespan (*Figure 2A''*). An example of increasing H3K9me2 enrichment across generations can be seen at multiple H3K9me2 peaks over a region of Chromosome III (*Figure 3—figure supplement 1A*). We also observed the same trend at the *rsks-1* locus (*Figure 3—figure supplement 1B*), the downregulation of which has been implicated in the increased lifespan of COMPASS mutants (*Han et al., 2017*). Overall, as lifespan increases across generations in *wdr-5* mutants compared to wild type (early- to mid- to late-gen) (*Figure 1*), we observed a corresponding increase in the ratio of mutant to wild-type coverage at each peak (*Figure 3C*, p<0.0001, paired t-test).

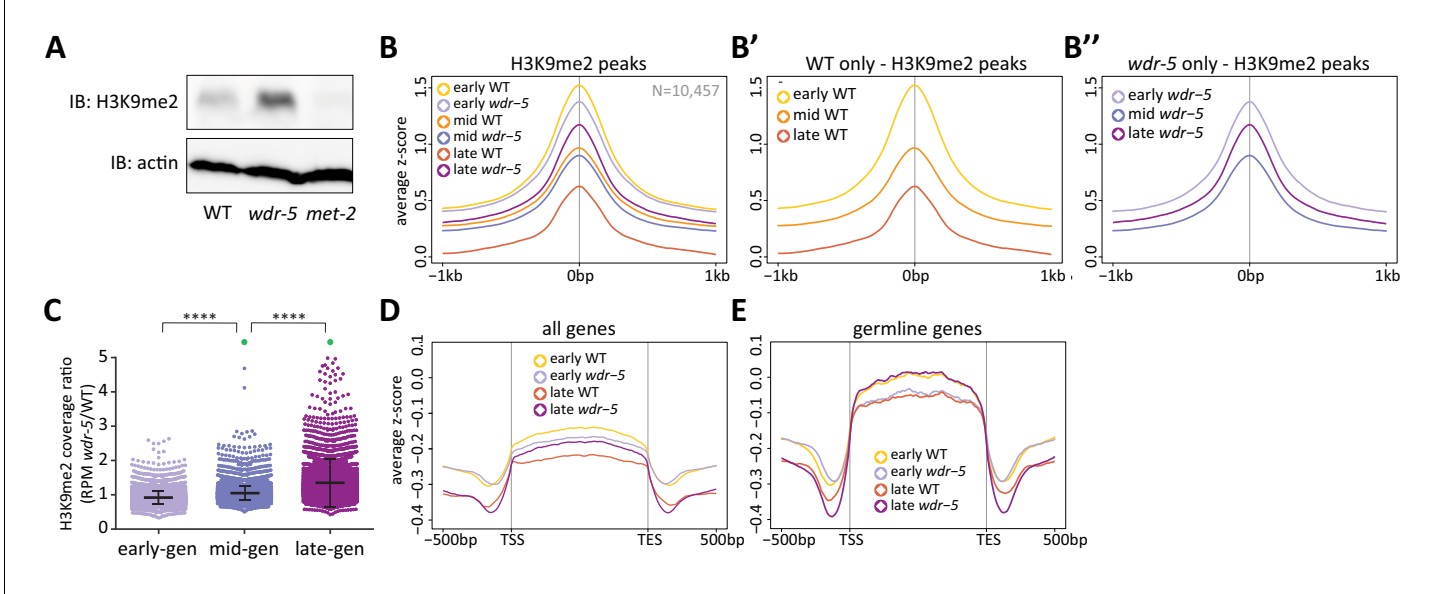

**Figure 3.** Long-lived *wdr-5* mutants have more H3K9me2 enrichment than wild type. (A) Immunoblot comparing H3K9me2 protein levels in late-gen wild type to late-gen *wdr-5* mutants and late-gen *met-2* mutants (representative of two independent experiments). Actin is used as a loading control. (B, D, E) Metaplots of averaged z-score H3K9me2 ChIP-seq signal across H3K9me2 peaks (B), all genes (D), or germline genes (E) in early-, mid-, and late-gen populations of wild type (yellow, orange, and red, respectively) and *wdr-5* mutants (lavender, purple, and plum, respectively). Line shows mean ChIP-seq signal. Data in (B) are also shown separated into wild type (B') and *wdr-5* mutants (B''). Plots are centered on peak centers (B, B', B'') or pseudoscaled over genes to 1 kb with 500 bp borders on either side, indicated by vertical gray lines (D–E). (C) H3K9me2 ChIP-seq ratios of *wdr-5* mutant coverage over wild-type coverage at each H3K9me2 peak (N = 10,457). Coverage is normalized to RPM. Thick line shows mean and whiskers show standard deviation. Green dots represent peaks that fall beyond y-axis scale (two peaks in mid-gen and 33 peaks in late-gen). ****p<0.0001 with paired t-test.

The online version of this article includes the following figure supplement(s) for figure 3:

**Figure supplement 1.** Long-lived *wdr-5* mutants have more H3K9me2 than wild type.

---

Because transgenerational phenotypes must be inherited through the germline, we would expect that H3K9me2 is most affected at germline-expressed genes in *wdr-5* mutants. To address this possibility, we examined H3K9me2 at germline-expressed genes (hereafter referred to as germline genes), including those expressed exclusively in the germline and those that are expressed in both the germline and soma (*Reinke et al., 2000*). At all genes, H3K9me2 enrichment was low and decreased from early- to late-gen in both wild-type and *wdr-5* mutant populations (*Figure 3D*). Likewise, at germline genes, wild-type H3K9me2 enrichment decreased between early- and late-gen populations (*Figure 3E*). In contrast, in *wdr-5* mutants, H3K9me2 enrichment at germline genes increased between early- and late-gen populations (*Figure 3E*). This increase countered the slight global decrease observed in *wdr-5* mutants between early- and late-gen populations (*Figure 3B''*). Thus, the general global retention of H3K9me2 became even more pronounced at germline genes.

## Requirement of MET-2 for *wdr-5* mutant longevity

Across generational time, higher levels of H3K9me2 generally correlated with a longer lifespan, raising the possibility that H3K9me2 helps to extend lifespan. To determine whether H3K9me2 is required for increased lifespan in *wdr-5* mutants, we examined animals lacking the H3K9 methyltransferase MET-2. *met-2* mutants have nearly undetectable amounts of H3K9me2 by immunoblot (*Figure 4A*), and have reduced H3K9me2 by mass spectrometry (*Towbin et al., 2012*), immunofluorescence (*Bessler et al., 2010*), and ChIP followed by quantitative PCR (ChIP-qPCR) (*Kerr et al., 2014*). Generally, *met-2* mutants were shorter-lived than wild-type populations, with lifespans on average 14% shorter (*Figure 4B*, p=0.0002, log-rank test, and *supplementary file 3*). If H3K9me2 is required for increased lifespan in *wdr-5* mutants, we would expect *wdr-5* mutants lacking H3K9me2 to have a short lifespan like *met-2* mutants. Furthermore, in the continued absence of H3K9me2, they should never be able to acquire longevity, even after many generations without WDR-5. To test

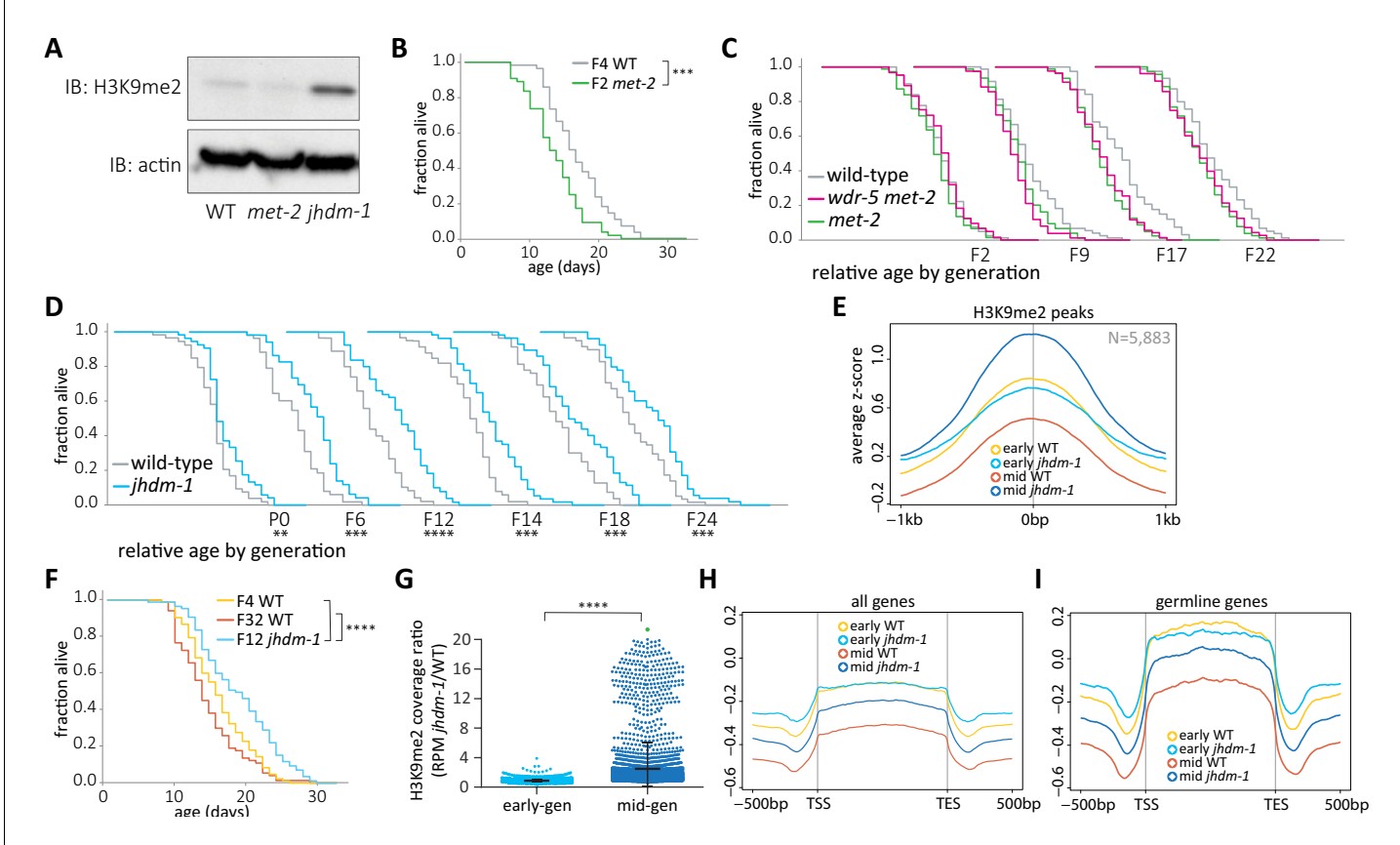

**Figure 4.** *wdr-5* mutant lifespan extension requires H3K9me2. (A) Immunoblot comparing H3K9me2 protein levels in mid-gen mixed-stage wild type to mid-gen *jhdm-1* and mid-gen *met-2* mutants (representative of two independent experiments). Actin is used as a loading control. (B, F) Lifespan of early-gen *met-2* mutants (green) and wild type (gray) (B) or mid-gen *jhdm-1* mutants (blue) compared to early- (yellow) and late-gen (orange) wild type (F). (C–D) Generational analysis comparing relative lifespan in *wdr-5 met-2* double mutants (pink) and *met-2* single mutants (green) (C) or *jhdm-1* mutants (blue) (D) to late-gen wild type (gray). The generation below each assay refers only to mutant populations. For each generation, the x-axis is set at is 40 days. (E, H, I) Metaplots of averaged z-score H3K9me2 ChIP-seq signal across H3K9me2 peaks (E), all genes (H), or germline genes (I) in early- and mid-gen populations of wild type (yellow and orange, respectively) and *jhdm-1* mutants (blue and navy, respectively). Line shows mean ChIP-seq signal. Plots are either centered on peak centers (E) or pseudoscaled over genes to 1 kb with 500 bp borders on either side, indicated by vertical gray lines (H– I). (G) H3K9me2 ChIP-seq ratios of *jhdm-1* mutant coverage over wild-type coverage at each H3K9me2 peak. Coverage for each sample is normalized to RPM. Thick line shows mean and whiskers show standard deviation. Green dot represents 65 peaks that lie beyond y-axis scale. *p<0.05, **p<0.01, ***p<0.001, and ****p<0.0001 compared to wild-type with log-rank test for lifespan assays or with paired t-test for coverage ratios. Median lifespan and statistics are presented in *supplementary file 1*, with additional replicates included in *supplementary file 3*.
The online version of this article includes the following figure supplement(s) for figure 4:

**Figure supplement 1.** Lifespan does not differ between populations descended from the same outcross.

these possibilities, we generated *wdr-5 met-2* double mutants and followed them for more than twenty generations, assessing lifespan periodically. The lifespan of *wdr-5 met-2* double mutants consistently resembled that of short-lived *met-2* mutants, and *wdr-5 met-2* mutants never became long-lived (*Figure 4C*).

## Increased H3K9me2 and lifespan in *jhdm-1* mutants

To determine whether higher levels of H3K9me2 can confer longevity directly, rather than as a consequence of changes in H3K4me, we examined mutant animals lacking a predicted H3K9 demethylase, JHDM-1. JHDM-1 has homology to *S. pombe* Epe1, a putative demethylase that prevents the inheritance of H3K9me2 across cell divisions (*Audergon et al., 2015*; *Ragunathan et al., 2015*). Consistent with a role in removing H3K9me, *jhdm-1* mutants have higher levels of global H3K9me2 compared to mid-gen wild-type animals by immunoblot (*Figure 4A*). To investigate this increase

across the genome, we next examined genomic H3K9me2 and compared it to lifespan in *jhdm-1* mutant populations across generational time. To avoid the complication of wild-type lifespan decreasing after a thaw, *jhdm-1* mutants were compared to late-gen wild-type populations that had already attained a steady-state lifespan. Importantly, lifespan in these late-gen wild-type populations was indistinguishable from that of wild-type populations derived from an outcrossed population that had been maintained as *jhdm-1/+* heterozygotes for five generations (p=0.878, log-rank test, *Figure 4—figure supplement 1*). Early-gen *jhdm-1* mutant populations were slightly longer lived than wild type (*Figure 4D*), and had similar levels of H3K9me2 enrichment at ChIP-seq peaks (*Figure 4E*, p=0.004, log-rank test). By mid generations, *jhdm-1* mutants had significantly longer lifespans than their wild-type counterparts, despite having no difference in progeny number (*Figure 4D*, p=0.0004, log-rank test, and *supplementary file 2*). With lifespans averaging 30% longer than wild type, *jhdm-1* mutants experienced a more robust longevity effect than *wdr-5* mutants (*supplementary files 1* and *3*). Similar to *wdr-5* mutants, the increase in *jhdm-1* mutant lifespan corresponded with a genome-wide increase in H3K9me2 enrichment at peaks (*Figure 4E*). Mid-gen *jhdm-1* mutants had more H3K9me2 than either early-gen *jhdm-1* mutants or any generation of wild type (*Figure 4E*). The increase in mid-gen *jhdm-1* mutants was particularly pronounced when examining RPM coverage ratios between *jhdm-1* mutants and wild type in mid- versus early-gen populations (*Figure 4G*, p<0.0001, unpaired t-test).

75% of peaks called in the *jhdm-1* mutant transgenerational experiment were shared with those called in the *wdr-5* mutant transgenerational experiment (*Figure 3—figure supplement 1C*). This overlap indicated that the H3K9me2 increases in both mutants occurred at similar locations in the genome. To further investigate the location of H3K9me2 peaks, we also examined coverage over all genes as well as over germline genes specifically in *jhdm-1* mutants. In early-gen populations, wild type and *jhdm-1* have similar levels of H3K9me2 at all genes (*Figure 4H*), including germline genes (*Figure 4I*). Across generations, wild type experienced a decrease in H3K9me2 enrichment at germline genes and this decrease is dependent upon WDR-5 (*Figure 3E*). Since the COMPASS complex should be functional in *jhdm-1* mutants, we would not expect H3K9me2 to be protected at germline genes across generations, as it was in *wdr-5* mutants. In long-lived mid-gen *jhdm-1* mutants, we did indeed observe less H3K9me2 enrichment at germline genes (*Figure 4I*), as we did at all genes (*Figure 4H*), although the reduction was not as large as we observed in their wild-type counterparts (*Figure 4I*). This H3K9me2 decrease over genes was particularly notable when compared to the overall accumulation of H3K9me2 at all peaks in mid-gen *jhdm-1* mutants (*Figure 4E*).

## Inheritance of longevity requires H3K9me2

Greer and colleagues found that genetically wild-type descendants of long-lived *wdr-5* mutants are as long-lived as their mutant ancestors for up to four generations (*Greer et al., 2011*). Since H3K9me2 is required for the lifespan extension of *wdr-5* mutants, it may also be the transgenerational factor inherited by their wild-type descendants. Using long-lived late-gen *wdr-5* mutants, we recapitulated the observation that F3 genetically wild-type descendants of *wdr-5* mutants (labeled WT (*wdr-5*)) were as long-lived as their *wdr-5* mutant cousins descended from the same population (labeled *wdr-5* (*wdr-5*)) (*Figure 5A and B*, p=0.04 and p=0.01, respectively, log-rank test) (*Greer et al., 2011*). As originally reported, we found that the lifespan extension of genetically wild-type descendants reverted by generation F5 (*Figure 5C*, p=0.47, log-rank test) (*Greer et al., 2011*). We next tested whether H3K9me2 is required for the inheritance of longevity by removing MET-2 from otherwise wild-type descendants of long-lived *wdr-5* mutants (*Figure 5D*). We used long-lived late-gen *wdr-5* mutants to generate homozygous *wdr-5* mutants that were also *met-2/+* heterozygote mutants (labeled '*wdr-5*'). F3 *met-2* mutant descendants of '*wdr-5*' mutants (labeled *met-2* ('*wdr-5*')) were significantly shorter-lived than a wild-type control population (17%, p<0.0001, log-rank test), and resembled the short lifespan of *met-2* mutants (*Figure 5—figure supplement 1*, p=0.48, log-rank test). In contrast, their F3 genetically wild-type cousins descended from the same ancestral population (labeled WT ('*wdr-5*')) were still long-lived (*Figure 5E*, p=0.04, log-rank test). Therefore, MET-2 was required for the inheritance of longevity by descendants of *wdr-5* mutants.

Eliminating MET-2 abrogated the inheritance of *wdr-5* mutant longevity, but this suppression could have been caused either by loss of H3K9me2 or by another factor responding to its absence. *met-2* mutants had shorter lifespans than wild type (*Figure 4B*). If H3K9me2 is the primary factor mediating inheritance of longevity, a normal chromatin state should be established by restoring

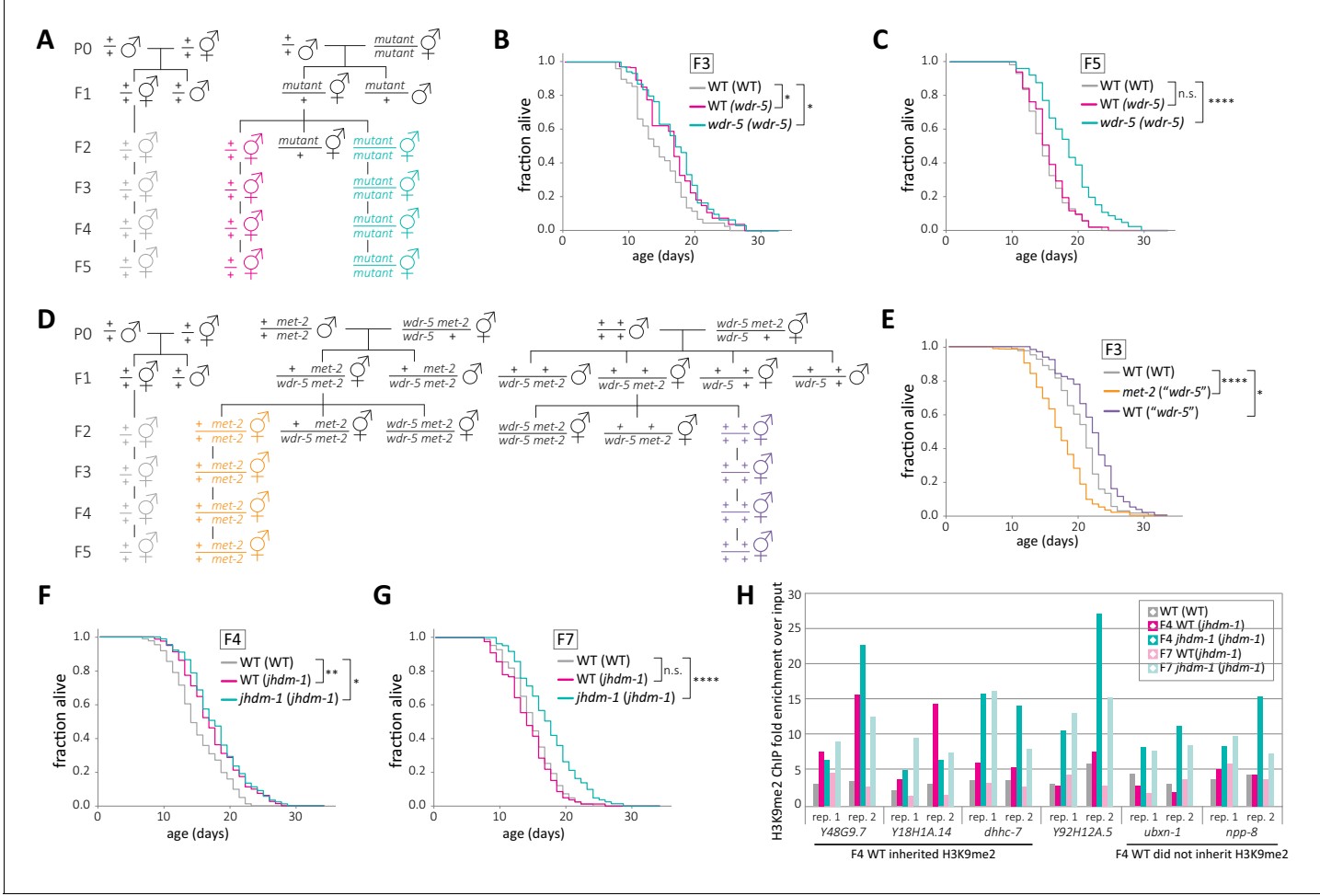

**Figure 5.** H3K9me2 facilitates the inheritance of longevity by wild-type descendants. (A, D) Genetic schemes for generating wild-type descendants from single mutants (A) or *met-2* mutant descendants from long-lived *wdr-5 met-2/wdr-5 +* mutants (D). (B–C) Lifespan of genetically wild-type F3 (B) or F5 (C) descendants of long-lived *wdr-5* mutants (WT (*wdr-5*), pink), compared to *wdr-5* mutants (*wdr-5* (*wdr-5*), teal)) and descendants of wild-type animals (WT (WT), gray). (E) Lifespan of F3 *met-2* mutants (*met-2* ('*wdr-5*'), gold) and genetically wild-type mutants (WT ('*wdr-5*'), purple)) descended from long-lived *wdr-5 met-2/+ wdr-5* mutants compared to descendants of wild-type animals (WT (WT), gray). (F–G) Lifespan of genetically wild-type F3 (F) or F5 (G) descendants of long-lived *jhdm-1* mutants (WT (*jhdm-1*), pink), compared to *jhdm-1* mutants (*jhdm-1* (*jhdm-1*), teal) and descendants of wild-type animals (WT (WT), gray). *p<0.05, **p<0.01 ****p<0.0001 compared to WT (WT) using log-rank test. Median lifespan and statistics are presented in *supplementary file 1*, with additional replicates included in *supplementary file 3*. (H) Fold enrichment of H3K9me2 ChIP compared to input measured by qPCR at six peaks enriched for H3K9me2 in mid-gen *jhdm-1* ChIP-seq. Two ChIP replicates are shown. Samples were mixed-stage animals from wild-type descendants of wild-type animals (WT (WT), gray), F4 and F7 genetically wild-type descendants from mid-gen *jhdm-1* mutants (WT (*jhdm-1*), dark and light pink respectively), and F4 and F7 *jhdm-1* mutants descended from mid-gen *jhdm-1* mutants (*jhdm-1* (*jhdm-1*), dark and light teal, respectively). Relative DNA quantities used to calculate fold enrichment are included in *supplementary file 4*.

The online version of this article includes the following figure supplement(s) for figure 5:

**Figure supplement 1.** The short lifespan of *met-2* mutants is not heritable.

**Figure supplement 2.** Summary figure of model.

H3K9me2 through the reintroduction of MET-2. Therefore, the decreased lifespan of *met-2* mutants should not be inherited by wild-type progeny. F3 and F5 genetically wild-type descendants of *met-2* mutants had a normal lifespan (*Figure 5—figure supplement 1*, p=0.34, log-rank test), consistent with H3K9me2 mediating longevity.

Similar to *wdr-5* mutants, *jhdm-1* mutants had longer lifespans and more H3K9me2 (*Figure 4*). If elevated H3K9me2 can directly mediate the transgenerational inheritance of longevity, we would also expect the longevity of *jhdm-1* mutants to be heritable. We examined the lifespan of genetically wild-type animals descended from long-lived *jhdm-1*. F4 wild-type animals descended from *jhdm-1*

mutants (labeled WT (*jhdm-1*)) had a significant increase in lifespan compared to wild-type controls (17%, p=0.007, log-rank test), and were as long-lived as F4 *jhdm-1* mutants descended from the same *jhdm-1* mutant population (labeled *jhdm-1* (*jhdm-1*)) (17% compared to wild-type controls, p=0.010, log-rank test) (*Figure 5F*, following the same genetic scheme as *Figure 5A*). By generation F7, the lifespan of genetically wild-type descendants from *jhdm-1* mutants reverted to resemble that of wild-type controls (*Figure 5G*, p=0.45). Both the inheritance of longevity and its reversion after five to seven generations mirrored what was observed in *wdr-5* mutants, indicating that the same mechanism may be responsible for both phenomena.

If H3K9me2 is the heritable factor that enables increased lifespan in wild-type descendants of long-lived *wdr-5* and *jhdm-1* mutants, we would expect H3K9me2 levels to remain high at some loci in long-lived wild-type descendants. To examine this possibility, we assayed H3K9me2 enrichment over generational time. By western blot, global H3K9me2 levels were fairly low, making it difficult to detect the inheritance of enriched H3K9me2. For a more quantitative measure of H3K9me2 enrichment, we used ChIP-qPCR on wild-type descendants of long-lived *jhdm-1* mutants, choosing six loci that had high levels of H3K9me2 in long-lived *jhdm-1* mutants by ChIP-seq (*Figure 5H*). At all loci, H3K9me2 levels were elevated in *jhdm-1* mutant populations and were relatively low in a late-gen wild-type control, which confirmed our ChIP-seq results (*Figure 5H*). At three of the loci examined, H3K9me2 levels remained high in long-lived F4 wild-type animals descended from long-lived *jhdm-1* mutants, and dropped back to low levels in F7 wild-type descendants of *jhdm-1* mutants. At a fourth locus, H3K9me2 levels remained high in one of two long-lived F4 wild-type replicates. At two other loci, H3K9me2 levels were low in all wild-type descendants of *jhdm-1* mutants, indicating that the elevated H3K9me2 was not inherited at a subset of loci.

## Discussion

Lifespan is a complex trait determined by many factors, including overall chromatin state. In this study, we find that lifespan gradually increases with each generation in populations lacking WDR-5 activity for at least twenty generations. These data provide a rare example of a complex trait acquired transgenerationally across successive generations.

We were surprised to find that wild-type lifespan decreases over generational time in populations recovering from a thaw. The changes in wild-type lifespan indicate that lifespan can be modulated transgenerationally irrespective of any mutations in chromatin modifiers. If a trait as complex as lifespan can change over time without a mutation, it seems likely that other phenotypes may be epigenetically regulated over generational time. In addition, the transgenerational nature of wild-type lifespan could account for some discrepancies in the *C. elegans* aging field. We find that after starving or recovering from a thaw, wild-type lifespan decreases for about twelve generations before reaching a steady state. These data, along with other studies, suggest that starving increases lifespan by altering the epigenetic landscape (*Rechavi et al., 2014*; *Webster et al., 2018*). Independent of any lifespan changes in wild-type populations, the extension of lifespan in *wdr-5* mutants is truly caused by loss of COMPASS activity, because late-gen *wdr-5* mutants live longer than any generation of wild type. This could account for why the transgenerational nature of *wdr-5* mutant longevity had not previously been reported. Ultimately, we see the greatest difference in lifespan when comparing late-gen wild-type and late-gen *wdr-5* mutant populations (on average, a 22% increase), recapitulating prior observations (*Greer et al., 2010*).

Two pieces of evidence indicate that the transgenerational increase of lifespan in *wdr-5* mutants is an epigenetic phenomenon. First, the COMPASS complex deposits transcription-coupled H3K4me, a modification that is able to be inherited between generations (*Heard and Martienssen, 2014*; *Katz et al., 2009*). Second, the lifespan increase in *wdr-5* mutants can be reset and subsequently reproduced, making it unlikely that longevity is caused by background genetic effects. But what changes over so many generations in *wdr-5* mutants? In the continued absence of COMPASS activity, *wdr-5* mutants have less H3K4me2/3 (*Greer et al., 2010*; *Li and Kelly, 2011*). However, it is difficult to imagine how a modification missing for so many generations could account for the inheritance of longevity in animals that have normal COMPASS activity (*Greer et al., 2011*). Thus, we wondered whether the loss of COMPASS activity affects another heritable factor.

The gradual appearance of longevity in *wdr-5* mutant populations suggests that this factor accumulates over many generations, while two other findings indicate that it resides in the germline.

First, heritable factors must be passed through germline tissue, whether they are genetically or epigenetically inherited. Second, a proliferating germline is required for longevity in *wdr-5* mutants (*Greer et al., 2010*). Therefore, in this study we examined whether H3K9me2 might be the heritable factor that confers longevity. Across generational time, we found that the relative enrichment of H3K9me2 generally correlates with longevity in both wild-type and *wdr-5* mutant populations, raising the possibility that H3K9me2 may be protective for lifespan. Our approach to ChIP-seq does not allow us to compare absolute quantities of H3K9me2 across samples, but we are able to detect a relative H3K9me2 enrichment in the populations that experienced the most significant changes in lifespan. For example, the largest decrease in wild-type lifespan occurs between early- and mid-gen populations, which is when we observe a genome-wide decrease in H3K9me2. Similarly, the largest increase in *wdr-5* mutant lifespan occurs between mid- and late-gen populations, correlating with a genome-wide increase H3K9me2 enrichment. Overall, among the transgenerational populations examined, we find that relative H3K9me2 enrichment changes most during the same generations that lifespan changes. However, it should be noted that, although the overall trend of H3K9me2 enrichment does parallel the lifespan changes we observe, the correlation is not absolute. For example, there are generational time periods where H3K9me2 enrichment decreases even as lifespan remains unchanged: in mid- to late-gen wild-type populations, and in early- to mid-gen *wdr-5* mutants. This discrepancy may be due to the fact that only a subset of loci, of the thousands identified, may be responsible for extending lifespan.

The general correlation between H3K9me2 enrichment and longevity is further supported by our finding that the H3K9 methyltransferase MET-2 is necessary both for a normal lifespan and for the lifespan extension of *wdr-5* mutants. However, multiple pathways are able to shorten lifespan (*Kenyon, 2010*). To test whether higher levels of H3K9me2 can extend lifespan independent of COMPASS activity, we examined animals lacking JHDM-1, a putative H3K9 demethylase. The absence of JHDM-1 is sufficient to increase lifespan. Additionally, longevity appears more quickly in *jhdm-1* mutants than in *wdr-5* mutants, perhaps because JHDM-1 has a more direct effect on H3K9me2 than COMPASS. Within a few generations, we observe that long-lived *jhdm-1* mutants have already experienced a large increase in H3K9me2 enrichment. This result provides further evidence of the correlation between longevity and H3K9me2. In addition, the majority of H3K9me2 peaks in long-lived *jhdm-1* mutants are shared with long-lived *wdr-5* mutants, which is consistent with the possibility that lifespan extension is caused by similar means in both mutants. Finally, we noticed that mid- or late-gen *jhdm-1* mutants appear healthier than their *wdr-5* mutant counterparts, indicating that H3K9me2 may affect health-span in addition to lifespan.

Based on our data, we developed the following model. Normally, WDR-5 functions as a component of COMPASS to add H3K4me during transcription. After thawing or starving, the genome has elevated H3K9me2, which is eroded in subsequent generations by the deposition of transcription-coupled H3K4me. This H3K9me2 reduction gradually reduces lifespan (*Figure 5—figure supplement 2A*). In wild type, the continued presence of COMPASS activity in the germline keeps H3K9me2 levels low over genes by maintaining H3K4me levels. COMPASS's maintenance of H3K4me over expressed genes occurs even when H3K9me2 levels are elevated overall in *jhdm-1* mutants. In *wdr-5* mutants, the lack of transcription-coupled H3K4me protects H3K9me2 in the genome, allowing it to accumulate at germline genes over generational time and extend lifespan (*Figure 5—figure supplement 2B*). Thus, we propose that the transgenerational inheritance of H3K9me2 extends lifespan in *wdr-5* mutants.

Our model may also explain how longevity is transgenerationally inherited in genetically wild-type descendants of COMPASS mutants (*Greer et al., 2011*). Inappropriately high H3K9me2 in long-lived *wdr-5* mutants could be inherited by their descendants, initially preventing COMPASS from restoring a steady-state chromatin environment and conferring longevity. After several generations of normal COMPASS activity, H3K4me would then reestablish wild-type levels of H3K9me2, resulting in the reversion to normal lifespan by the fifth generation (*Figure 5—figure supplement 2B*). Consistent with H3K9me2's involvement in the inheritance of longevity, we find that MET-2 is necessary for the inheritance of *wdr-5* mutant longevity. Furthermore, if H3K9me2 mediates inheritance directly, we would not expect the short lifespan of *met-2* mutants to be inherited when H3K9me2 is restored by MET-2. We show that F3 genetically wild-type descendants of *met-2* mutants have normal lifespans, suggesting that H3K9me2 is directly involved in inheritance of longevity. However, this genetic

interaction does not distinguish whether MET-2's effect on inheritance is directly through H3K9me2 or through another factor responding to the absence of H3K9me2.

The extended lifespan of *jhdm-1* mutants allows for an independent test of whether H3K9me2 mediates the inheritance of longevity. If H3K9me2 is the mechanism by which descendants of COMPASS inherit their longevity, it should be heritable and confer longevity no matter the genetic background. Similar to what was previously reported for *wdr-5* mutants, we find that F4 wild-type descendants of long-lived *jhdm-1* mutants are long-lived, and longevity reverts to normal in F5-F7 descendants. In addition, at four out of six H3K9me2 ChIP-seq peaks examined by ChIP-qPCR, we find evidence that elevated H3K9me2 levels remain high in F4 genetically wild-type descendants of long-lived *jhdm-1* mutants. This result suggests that H3K9me2 enrichment can be inherited at a subset of loci. By F7, when the lifespan of genetically wild-type descendants has reverted to wild-type levels, we find that H3K9me2 enrichment has also decreased back to wild-type levels at these four loci. Based on these observations, we propose that H3K9me2 is also the factor that allows for the transgenerational inheritance of longevity.

Previously, mutants lacking the H3K4 demethylase SPR-5 were found to acquire increased lifespan after six to ten generations (*Greer et al., 2016*). Thus, we considered the possibility that *spr-5* and *wdr-5* mutants share a common mechanism. However, *spr-5* mutants accumulate the active modification H3K4me2 (*Katz et al., 2009*), which differs from the H3K9me2 accumulation observed in *wdr-5* mutants. Additionally, *spr-5* mutant longevity is mediated by the known DAF-36/DAF-12 lifespan signaling pathway (*Greer et al., 2016*), whereas the lifespan extension of COMPASS mutants occurs independent of this pathway. Therefore, it is likely that different mechanisms underlie the transgenerational acquisition of longevity in these mutants.

Although this work demonstrates that H3K9me2 functions in the establishment and inheritance of extended lifespan, it does not address the mechanism by which H3K9me2 affects longevity. However, the increased lifespan in COMPASS mutants has been linked to the fatty acid desaturation pathway in the intestine (*Han et al., 2017*). It is not clear how these metabolic changes correlate with the transgenerational acquisition of longevity in COMPASS mutants or its inheritance by wild-type descendants. Nevertheless, the increase in mono-unsaturated fatty acids in COMPASS mutants is at least partially mediated by downregulating germline target genes, including the S6 kinase *rsks-1* (*Han et al., 2017*). When we examined H3K9me2 at this locus over generational time, we found that late-gen *wdr-5* mutants had much higher H3K9me2 enrichment than their wild-type counterparts, matching the overall trend we see at all H3K9me2 peaks. Thus, it is possible that the accumulation of mono-unsaturated fatty acids in COMPASS mutants may be caused by higher levels of H3K9me2 in long-lived populations. Further work will be necessary to determine whether this is the case.

Overall, we have established a role for a heterochromatic histone modification in both the establishment and inheritance of a complex trait. This work also relates two seemingly disparate theories of aging that are widely observed among eukaryotes: the disposable soma theory of aging, which is based on observations that reproductive ability often comes at the expense of lifespan (*Kirkwood and Holliday, 1979*), and the heterochromatin loss model of aging, which is based on the observation that heterochromatin declines with age (*Tsurumi and Li, 2012*; *Villeponteau, 1997*). Our model proposes a mechanism that could connect both theories: if reduced heterochromatin is a burden on lifespan, then limiting H3K9me2 through transcription-coupled H3K4me deposition in the germline may represent one cost of maintaining a germline.

## Materials and methods

### Key resources table

| Reagent type (species) or resource | Designation | Source or reference | Identifiers | Additional information |
|---|---|---|---|---|
| Genetic reagent (*Caenorhabditis elegans*, hermaphrodite) | N2 wild type | Wormbase | WB Cat# N2_(ancestral), RRID:WB-STRAIN:N2_(ancestral) | |

*Continued on next page*

*Continued*

| Reagent type (species) or resource | Designation | Source or reference | Identifiers | Additional information |
|---|---|---|---|---|
| Genetic reagent (*C. elegans*, hermaphrodite) | *wdr-5 (ok1417) III* | Wormbase | WB Cat# RB1304; RRID:WB-STRAIN:RB1304 | |
| Genetic reagent (*C. elegans*, hermaphrodite) | *met-2 (n4256) III* | Wormbase | WB Cat# MT13293; RRID:WB-STRAIN:MT13293 | |
| Genetic reagent (*C. elegans*, hermaphrodite) | *jhdm-1 (ok2364) III* | Wormbase | WB Cat# RB1826, RRID:WB-STRAIN:RB1826 | |
| Genetic reagent (*C. elegans*, hermaphrodite) | *wdr-5 (ok1417) met-2 (n4256)/qC1 qIs26 [lag-2::GFP + pRF4 rol-6(su1006)] III* | this paper | | recombinant chromosome III isolated by crossing MT13293 and RB1304, then crossed to KW2203 to maintain over qC1 balancer |
| Antibody | Mouse monoclonal anti-H3K9me2 | Abcam | Abcam:ab1220; RRID:AB_449854 | WB: 1:500 |
| Antibody | Mouse monoclonal anti-actin | Millipore/Upstate | Millipore:MAB1501; RRID:AB_2223041 | WB: 1:5000 |
| Antibody | rabbit polyclonal secondary anti-mouse IgG (HRP) | Abcam | Abcam:ab6728; RRID:AB_955440 | WB: 1:3000 |
| Commercial assay or kit | ECL Plus | Amersham Biosciences | Amersham Biosciences:RPN2106 | |
| Commercial assay or kit | Chromatin Immunoprecipitation Assay Kit | EMD Millipore | EMD Millipore:17–295 | |
| Commercial assay or kit | iO SYBR Green Supermix | Bio-Rad | Bio-Rad:1708882 | |
| Software | Bowtie2 | PMID: 22388286 | RRID:SCR_005476 | |
| Software | MACS v2.1.1 | PMID: 22936215 | RRID:SCR_013291 | |
| Software | deepTools2 | PMID: 27079975 | RRID:SCR_016366 | |
| Software | Integrated Genome Viewer (IGV) | PMID: 22517427 | | |
| Software | SeqPlots | PMID: 27918597 | | |
| Software | BEDtools v2.27.1 | PMID: 20110278 | RRID:SCR_006646 | |

## Strains

All *C. elegans* strains were cultured at 20°C on 6 cm nematode growth media (NGM) agar plates with OP50 bacteria grown in Luria Broth (LB). Strains used were: N2: wild-type (Bristol isolate); RB1304: *wdr-5 (ok1417) III*; MT13293: *met-2 (n4256) III*; RB1826: *jhdm-1 (ok2364) III*; and *wdr-5 (ok1417) met-2 (n4256)/qC1 qIs26 [lag-2::GFP + pRF4 rol-6(su1006)] III*.

## Single-worm genotyping

Single animals were picked into 5–10 μl of lysis buffer (50 mM KCl, 10 mM Tris-HCl (pH 8.3), 2.5 mM MgCl$_2$, 0.45% NP-40, 0.45% Tween-20, 0.01% gelatin) and incubated at 65°C for 1 hr followed by 95°C for 30 min. PCR reactions were performed with AmpliTaq Gold (Invitrogen) according to the manufacturer's protocol and reactions were resolved on agarose gels. The following genotyping primers were used: *wdr-5 (ok1417)*: CCCAAACTCCCAATCCAAACG, GTGTGCTGGGAGGGTTTTTA, GGATGACAATCGGAGGCTAG; *met-2 (n4256)*: GTCACATCACCTGCATCAGC, ATTTCATTACGGCTGCCAAC, ATTCGAAAAATGGACCGTTG, TCTATTCCCAGGAGCCAATG; *jhdm-1 (ok2364)*: GAAATAAATGCGTGCCGGACC, CGTTCTAGTTCAAGACGTTCAGGTG, TCCATTCTGGGATCATAGTTATACG.

## Transgenerational experiments

For each strain, three L4 hermaphrodites were transferred every fourth day from the previous population, except for *met-2* or late-generation *wdr-5* populations, in which four to six gravid young

adults were transferred to ensure the selection of fertile animals. To reset populations, strains were either thawed or maintained as heterozygote mutant populations for five generations before homozygous mutant animals were selected to be the P0 generation. For analysis of populations recovering from starvation, progeny of arrested L1s were considered P0. For analysis of populations recovering from a thaw, progeny of surviving L1s were considered P0. Populations were genotyped periodically throughout the transgenerational experiment. For the starvation experiment in *Figure 1—figure supplement 2C*, gravid hermaphrodites from late-gen populations were laid embryos on NGM plates with no OP50 bacterial lawn. Hatched L1s were kept without food for six days before being transferred to plates with OP50. The progeny of these starved L1s were used in the lifespan assay.

## Lifespan assays

Assays were performed at 20℃ on NGM agar plates that did not contain 5-fluoro-2'-deoxyuridine (FUdR). On Day 1, young adults (on their first day of egg-laying) were allowed to lay for 4–6 hr to hatch a synchronized population for the assay. When progeny were L4s or young adults, 90 animals per condition were transferred to new plates, with 30 animals per plate. Animals were transferred every day or every other day during their fertile period (usually the first ten days of adulthood). Plates were scored daily and animals marked as dead if they did not move in response to repeated prodding with a platinum pick. Animals were censored from analysis if they died from ruptured vulvas, matricide ('bag of worms' phenotype), or crawling off the agar. Kaplan-Meier survival curves were generated in GraphPad Prism and significance was calculated using a log rank test (Mantel-Cox). Lifespan differences are reported as percentages of median lifespans. The following core observations were repeated in double-blind experiments: transgenerational acquisition of longevity in *wdr-5* mutants (*Figure 1*), the change in wild-type lifespan after freezing (*Figure 2A*), and the requirement for MET-2 in inheritance of *wdr-5* mutant lifespan (*Figure 5E*).

## Progeny count assay

Individual hermaphrodites were cloned as L4s and transferred daily until no longer fertile. Progeny were scored as L4s or young adults. Each experiment started with broods of at least five animals, although broods were censored if mothers died before the end of their laying period. Significance was calculated using an unpaired t-test.

## Protein analysis by immunoblot

To generate protein extract, animals were cultured with OP50 on six to twelve 10 cm NGM agar plates. Mixed-stage populations were collected by washing off plates with PBS, pelleted in 500 µl of PBS, and flash frozen. Frozen pellets were thawed, resuspended in NE2 buffer (250 mM sucrose,10 mM HEPES (pH 7.9), 450 mM NaCl, 2 mM $MgCl_2$, 2 mM $CaCl_2$, 0.1% Triton-X100), and flash frozen. Frozen pellets were then disrupted by a 7 ml Type B glass Dounce homogenizer and allowed to lyse on ice for 15 min. Pellets were washed two times in cold PBS, then resuspended in 20 mM Tris-HCl (pH 7.9). Extracts were resolved with 12% Mini-PROTEAN TGX Stain-Free Protein Gels (BioRad) and transferred to nitrocellulose membranes. Primary antibodies were: 1:500 H3K9me2 antibody (ab1220, Abcam) and 1:5000 actin (MAB1501 (Millipore/Upstate)). Primary antibodies were visualized using 1:3000 Rabbit Anti-Mouse IgG H and L (HRP) (ab6728, Abcam) and ECL Plus (Amersham Biosciences). Quantification was performed using a ChemiDoc MP and Image Lab software (BioRad).

## Chromatin immunoprecipitation

To generate chromatin extract, animals were cultured with OP50 on six to twelve 10 cm NGM agar plates. Mixed-stage populations were collected by washing off plates with PBS, pelleted in 500 µl of PBS, and flash frozen. Frozen pellets were disrupted by a 7 ml Type B glass Dounce homogenizer, fixed for ten minutes with 1% formaldehyde (diluted from 37% (w/v)) at 37℃ and quenched with 125 mM glycine. ChIP samples were processed with a Chromatin Immunoprecipitation Assay Kit (EMD Millipore) according to manufacturer's instructions. Samples were sonicated using a Diagenode Bioruptor UCD-200 at 4℃ for 15 min on high, with a cycle of 45 s on and 15 s off. 1/20th of sample volume was taken for input controls. For immunoprecipitation, extracts were incubated overnight at 4℃ with 10 µl of H3K9me2 antibody (ab1220, Abcam). DNA was extracted by phenol-chloroform and ethanol precipitated.

## High-throughput sequencing and data analysis

Samples were sent for library preparation and sequencing at either the HudsonAlpha Genomic Services (Huntsville, AL, USA) or the Georgia Genomics and Bioinformatics Core (Athens, GA, USA). For *wdr-5* mutant replicates 1 and 2 and for *jhdm-1* mutant replicate 1, single-end 50 bp sequencing was performed on the Illumina HiSeq v4 platform. For *wdr-5* replicate three and *jhdm-1* replicate 2, single-end 75 bp sequencing was performed on the Illumina NextSeq platform. Reads were filtered and aligned to genome WS220/ce10 using Bowtie2 (*Langmead and Salzberg, 2012*) using default settings. Peaks were called with MACS v2.1.1 (*Feng et al., 2012*) using the following parameters: –bw 150 -q. 01 –nomodel –broad. Peaks were required to be present in at least two replicates from the same genotype and generation, and the union of all peaks was used for the final peak list. bam-Coverage in deepTools2 (*Ramírez et al., 2016*) was used to generate bedGraph coverage tracks in 10 bp bins, with blacklisted regions excluded (blacklist obtained from *McMurchy et al., 2017*), using the following parameters: -bs 10, -e 200, –normalizeUsing None. Z-score was used to normalize coverage in 10 bp bins for each replicate, and the average for each bin was calculated from all three replicates to generate average z-score coverage tracks. bedGraphs were converted to bigWig coverage tracks using UCSC bedGraphToBigWig utility. Tracks were visualized on Integrated Genome Viewer (IGV) (*Thorvaldsdóttir et al., 2013*). Metaplots were visualized from average z-score coverage tracks using SeqPlots (*Stempor and Ahringer, 2016*). Coverage counts over peaks was identified using multicov in bedtools v2.27.1 (*Quinlan and Hall, 2010*), normalized to reads per million, and averaged between all three replicates for coverage ratio analysis. The list of germline genes was modified from *Reinke et al. (2004)*.

## Quantitative PCR

Samples were prepared for ChIP as above, with the addition of a no-antibody-added control. DNA from two replicate ChIPs was quantified by real-time PCR, using iO SYBR Green Supermix (BioRad). The following primers were used: *Y48G9.7* (ATCTGCTTGGGACACTGC and AAATTGGACGAC TGCAACAGC) *Y18H1A.14* (ATCAGTGAACACGGGATTCTGG and TTTGGCTCGGACATATCTGG); *C17D12.1* (CCTGAATCGTTCATCTGCAACTG and TTTCCTGACACAACGCTTGC); *Y92H12A.5* (GA TCCGCCAAGTGATCTACAGTC and ACCCATCGTCGCCTCACTAATAC); *ubxn-1* (AGAACGAA-GACGAAATCGCCAG and CAGGCTTTGCCTCTGGAACC); *npp-8* (GAGATTGGTGCAGAGTGCTG TG and AGCAGCTCTCAAGAGGCAAAG). Fold-enrichment was calculated as ((Ab ChIP/Ab input) / (no-Ab ChIP/no Ab input)). Raw relative DNA quantities are included in *supplementary file 4*.

## RNA interference

*Escherichia coli* HT115 transformed with a vector expressing dsRNA of *wdr-5.1* was obtained from the Ahringer library (Source BioScience). RNAi bacteria and an empty vector control was grown at 37C and seeded on RNAi plates (standard NGM plates containing ampicillin (100 mg/ml) and isopropylthiogalactoside (IPTG; 0.4 mM)) left at room temperature to induce for at least 24 hr. Gravid adults were placed on RNAi plates for six hours to obtain synchronized populations of worms. For lifespan assays, L4 animals obtained from these synchronized populations were transferred to fresh RNAi plates seeded with the respective bacteria; animals were transferred to freshly seeded RNAi plates every two days during their fertile period, and once a week thereafter. Transgenerational populations were maintained transferred to fresh RNAi plates every two days.

## Acknowledgements

We are grateful to W Kelly, M Mondoux, B Wheeler, and members of the Katz lab for their helpful discussion and critical reading of the manuscript; N Sharma, M J Rowley, and C Scharer for advice on bioinformatics; R Deal, V Corces, and W Kelly for the generous sharing of reagents and equipment; and V Reinke for her list of germline-expressed genes. We thank R Horvitz and the *Caenorhabditis* Genetics Center (funded by NIH P40 OD010440) for strains.

## Additional information

### Funding

| Funder | Grant reference number | Author |
|---|---|---|
| National Institutes of Health | K12GM00680-15 | Teresa W Lee<br>Brandon Scott Carpenter |
| National Science Foundation | IOS1354998 | David John Katz |
| National Institutes of Health | F31 NS098663-01A1 | Amanda Kathryn Engstrom |

The funders had no role in study design, data collection and interpretation, or the decision to submit the work for publication.

### Author contributions

Teresa Wei-sy Lee, Conceptualization, Formal analysis, Investigation, Visualization, Methodology, Project administration; Heidi Shira David, Formal analysis, Investigation; Amanda Kathryn Engstrom, Investigation; Brandon Scott Carpenter, Conceptualization; David John Katz, Conceptualization, Resources, Formal analysis, Supervision, Funding acquisition, Project administration

### Author ORCIDs

Teresa Wei-sy Lee (iD) https://orcid.org/0000-0002-0149-5389
Amanda Kathryn Engstrom (iD) https://orcid.org/0000-0002-6796-2752
Brandon Scott Carpenter (iD) https://orcid.org/0000-0001-9939-1926
David John Katz (iD) https://orcid.org/0000-0002-3040-1142

### Decision letter and Author response

Decision letter https://doi.org/10.7554/eLife.48498.sa1
Author response https://doi.org/10.7554/eLife.48498.sa2

## Additional files

### Supplementary files

• Supplementary file 1. Summary statistics for lifespan data in *Figures 1–5*. Shading indicates groupings of populations assessed in a single lifespan assay. Median lifespan was calculated from Kaplan-Meier survival curves and *P* values were calculated using a log-rank test *p<0.05, **p<0.01, ***p<0.001, ****p<0.0001. N indicates the number of observed dead animals at the end of the experiment, with the initial number of living animals indicated in parentheses. The difference corresponds to the number of individuals censored for deaths via matricide, vulval rupture, or desiccation from crawling off the plate. Figure panels for specific experiments are indicated in column 8. Replicate number is indicated in column 9.

• Supplementary file 2. Summary statistics for progeny number and embryonic lethality used in supplementary figure 2. Percent survival was calculated from counting the number of embryos laid and the number of surviving adults. Each experiment started with at least five broods; broods were censored if mother died by matricide or vulval rupture.

• Supplementary file 3. Summary statistics for lifespan data in figure supplements and additional replicates of lifespan assays. Shading indicates groupings of populations assessed in a single lifespan assay. Median lifespan was calculated from Kaplan-Meier survival curves and *P* values were calculated using a log-rank test *p<0.05, **p<0.01, ***p<0.001, ****p<0.0001. N indicates the number of observed dead animals at the end of the experiment, with the initial number of living animals indicated in parentheses. The difference corresponds to the number of individuals censored for deaths via matricide, vulval rupture, or desiccation from crawling off the plate. Figure panels for specific experiments are indicated in column 8. If data are not represented in a figure, the figure that shows its replicate is indicated. Replicate number is indicated in column 9.

- Supplementary file 4. Relative DNA quantifies of samples used to calculate fold enrichment by H3K9me2 ChIP-qPCR. Two biological replicates of H3K9me2 ChIP were conducted for each condition (each with three technical replicates). Error shown is standard deviation of the technical replicates. Fold enrichment was calculated using the following formula: ((Ab ChIP/Ab input) / (no-Ab ChIP/no Ab input)).

- Transparent reporting form

### Data availability

Sequencing data have been deposited in GEO under accession code GSE129928.

The following dataset was generated:

| Author(s) | Year | Dataset title | Dataset URL | Database and Identifier |
|---|---|---|---|---|
| Teresa Wei-sy Lee, Heidi Shira David, Amanda Kathryn Engstrom, Brandon Scott Carpenter, David John Katz | 2019 | H3K9me2 protects lifespan against the transgenerational burden of germline transcription in C. elegans | https://www.ncbi.nlm.nih.gov/geo/query/acc.cgi?acc=GSE129928 | NCBI Gene Expression Omnibus, GSE129928 |

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
