## [Decision Letter]

**Acceptance summary:**

There is growing recognition that longevity and healthspan can be impacted transgenerationally through epigenetic regulation. This has been best characterized in the nematode *Caenorhabditis elegans* through studies showing that genetic perturbation of histone methylation in prior generations can have far reaching effects on lifespan in future generations. Lee and colleagues provide new mechanistic insight into this phenomenon by identifying specific histone methylation marks that mediate transgenerational inheritance of lifespan for up to 20 generations.

**Decision letter after peer review:**

Thank you for submitting your article "H3K9me2 protects lifespan against the transgenerational burden of germline transcription in *C. elegans*" for consideration by *eLife*. Your article has been reviewed by three peer reviewers, one of whom is a member of our Board of Reviewing Editors, and the evaluation has been overseen by Kevin Struhl as the Senior Editor. The following individual involved in review of your submission has agreed to reveal their identity: Carlos Gio Silva Garcia (Reviewer #3).

The reviewers have discussed the reviews with one another and the Reviewing Editor has drafted this decision to help you prepare a revised submission.

Summary:

In this manuscript, Lee et al. hypothesized that the retention of H3K9me2 across generations correlates with the transgenerational longevity in the COMPASS complex mutant *wdr-5*, and is required for the inheritance of longevity to genetically wild-type offsprings descended from *wdr-5* homozygous mutants. The authors began with an observation that early generation *wdr-5* mutants do not live longer than early generation wild-type worms recovered from a thaw. However, *wdr-5* mutant lifespan lengthens over generations while wild-type lifespan shortens over generations. The authors correlated these lifespan changes with differences in H3K9me2. They observed that levels of H3K9me2 enrichment at ChIP-seq peaks and at gene bodies decrease over generations in wild-type worms while they are retained more effectively in *wdr-5* mutant worms. The authors then took genetic approaches to show that H3K9me2 is required for *wdr-5* longevity and its transgenerational inheritance. They further showed that the mutant for the putative histone demethylase, *jhdm-1*, has a similar pattern of H3K9me2 enrichment and longevity inheritance as *wdr-5* mutants.

In general, this study is interesting and very relevant to the field. If the authors' model was correct, it would represent a fundamental step forward towards understanding how chromatin modifiers can influence longevity across generations. Particularly interesting is the idea that H3K9me2 could be the inherited mark that confers longevity to COMPASS complex mutants, as this is an unresolved question in the field. The authors have clearly put much effort into these experiments, however, there are concerns regarding to some of the technical aspects of the experiments and the interpretations of some of the results which need to be addressed prior to publication.

Essential revisions:

1) Because the effects the authors are reporting are highly dependent on experimental technique – for example the reported changes in *wdr-5* lifespan occur over many generations in the lab – it is critically important that these experiments are well-controlled. The authors have generally done a good job of replicating their findings multiple times. However, it is unclear why they used N2 from frozen stock as their control when the *wdr-5* mutants came from a heterozygote cross. A more appropriate control would have been the wild-type siblings from the original *wdr-5* cross (outcrossed for more than 5 generations to eliminate the inherited longevity effects as in Figure 1). Those generation-matched wild-type siblings could have been compared to *wdr-5* mutants. Completing this experiment, at least for an early generation time-point, would validate the authors' observations that *wdr-5* mutants are not long-lived until later generations compared to generation-matched wild-types while avoiding the problem of recently-thawed worms which themselves have changes in longevity over time.

2) The author's model that *wdr-5* mutation only confers a longevity benefit after many generations or when compared to late-generation wild-type is not sufficient to explain the previous observation that *wdr-5* RNAi leads to increased longevity in worms within one generation (Greer et al., 2010). The authors should attempt to reproduce this finding, perhaps in early and late-generation wild-type worms, and comment on how their model can incorporate the RNAi phenotype of *wdr-5*. It is necessary to provide evidence that explains why the RNAi, in only one generation, is enough to extend lifespan meanwhile the *wdr-5* mutants, after outcross, do not have this phenotype in early-gen populations. The type of food (in the paper, animals were grown on OP50 bacteria and for the RNAi is on HT115 bacteria) or that *wdr-5* is in an operon (CEOP3136) could explain this difference.

3) Related to #1, the wild-type lifespans throughout the paper are extremely variable which make the data difficult to interpret. Although *C. elegans* wild-type lifespan is known to be variable, the typical range at 20°C is between 17-20 days, and the authors present several figures with average wild-type lifespans of 23-24 days (Figure 1—figure supplement 1), whereas other figures have much shorter average lifespans. The differences do not seem to be only due to the change in generation after a thaw, as in Figure 1 the mid-generation lifespans for wild-type are 23-24 days, at which time the authors showed in Figure 2 that wild-type lifespan should already have decreased. Given that the authors expect mid-generation worms to no-longer have this extended lifespan after recovering from a thaw, they should attempt to explain these abnormally high results.

4) In the authors' working model (Figure 5—figure supplement 2), they propose that H3K9me2 levels are decreasing over generations in wild-type animals but become stable or elevated in *wdr-5* mutants and their genetically wild-type F1-F4 descendants (Figure 5B—figure supplement 2). However, ChIP-seq experiments without a spike-in control do not allow direct quantitative comparison of H3K9me2 levels between samples. The metaplots of H3K9me2 ChIP-seq signal in Figure 3 and Figure 4 only indicate redistribution of H3K9me2 within a given sample but not changes in absolute quantity. To validate their model, the authors should perform ChIP-seq experiments with a spike-in control. Alternatively, they could also do Western blotting to measure the global levels of H3K9me2 in early/mid/late-generations of wild-type, *wdr-5*, and *jhdm-1* mutants and genetically wild-type descendants of *wdr-5* (F1/F3/F5 in Figure 5B—figure supplement 2).

5) The authors attempted to claim that H3K9me2 may be the inherited mark that accounts for the inheritance of longevity in wild-type offspring of COMPASS complex mutants. They said that H3K4 is unlikely to be the inherited mark, which is presumably based on evidence from the Greer et al., 2011 paper that shows that wild-type descendants of *wdr-5* mutants regain their H3K4me3 global levels even though they are still long-lived. However, if the authors want to claim that H3K9me2 may be this inherited mark, they need to show that unlike H3K4me3, H3K9me2 levels do not return to normal in wild-type offspring of *wdr-5* and *jhdm-1* mutants. This is a critical experiment and would make their paper much more convincing. Specifically, the authors should immunoblot for H3K9me2 (and H3K4me3 to confirm previously published work) in *wdr-5* and *jhdm-1* mutants, wild-type long-lived offspring, and normal wild-type worms as suggested in the previous comment.

6) It seems like the authors try to make a correlation between levels of H3K9me2 enrichment and lifespan phenotype. However, this explanation does not always fit the data. For example, wild-type, *wdr-5*, and *jhdm-2* animals all have decreased H3K9me2 enrichment at ChIP-seq peaks and from early to late generations. However, *wdr-5* and *jhdm-2* become long-lived at late-generations while wild-type worms have shortened lifespan. The authors should attempt to reconcile these patterns more clearly in their Discussion and avoid presenting the data as clear-cut. Additionally, it could be helpful to do a more gene-specific analysis, as perhaps the authors are missing the most important regions by looking at the average plots of all peaks, or even all germline genes. For example, the authors could look at differential peaks between wild-type and *wdr-5* mutants, and check for functional categories of the nearest genes to break down this analysis further in an informative way.

7) Figure 4D is difficult to interpret because all wild-type worms used were late-generation according to the supplementary table. If the authors want to claim there is a gain in *jhdm-1* lifespan similarly to *wdr-5* lifespan over generations, they should repeat this experiment with the appropriate generation-matched controls. Alternatively, they should make it very clear in the text that they used late-generation wild-type animals, as it currently reads as if there were generation-matched controls.

8) Figure 5E-G is also difficult to interpret because the authors are missing several important controls. The WT(WT) used does not appear to be a descendent from a control cross as was done for Figure 5B and C, so it is unclear whether the WT(WT) lifespan is really a proper control. Additionally, for Figure 5E there is no *met-2* mutant control for lifespan. Since the authors claim that *met-2* mutants are usually short-lived, there should be a *met-2* mutant control to test whether being descended from a *wdr-5* mutant could allow for increased lifespan relative to the shortened *met-2* lifespan, not just wild-type lifespan. Additionally, the only replicate for Figure 5F shows no statistically significant difference between WT(*jhdm-1*) lifespan and WT lifespan, making the authors' claims in Figure 5F somewhat weak. Adding an additional replicate would strengthen this claim. Importantly, there appears to be no replicate of Figure 5G. In order to keep Figure 5F and G, an additional replicate needs to be added to Figure 5G at the minimum.

9) *jhdm-1* is predicted to specifically demethylate H3K36 instead of H3K9me2. While it is interesting that H3K9me2 enrichment patterns change in *jhdm-1*, it could be an indirect consequence of the alterations of H3K36 methylation. Moreover, *jmjd-1.2* is the putative H3K9/H3K27 demethylase. The authors should include the *jmjd-1.2* mutants in the genetic analysis to rule out that *jmjd-1.2* is the more relevant demethylase.

10) The model that the WDR-5/MET-2/JHDM-1 pathway induces transgenerational inheritance of longevity through germline could be strengthened. Western blot of H3K9me2 in dissected gonads of different backgrounds/conditions and a germline rescue for *met-2* in the *wdr-5* background across generations will help this.

[Editors' note: further revisions were requested prior to acceptance, as described below.]

Thank you for resubmitting your work entitled "Repressive H3K9me2 protects lifespan against the transgenerational burden of COMPASS activity in *C. elegans*" for further consideration by *eLife*. Your revised article has been evaluated by Kevin Struhl (Senior Editor) and a Reviewing Editor.

The manuscript has been improved but there are some remaining issues that need to be addressed before acceptance, as outlined below:

1) The *wdr-5.1* RNAi data should be included as supplementary data. An obvious question that the readers would have after reading about the *wdr-5* mutant data would be how they could be reconciled with the previously published RNAi data. The authors should include their own *wdr-5* RNAi data, to indicate that at least under their experimental condition, the RNAi data are not inconsistent with their mutant data.

2) There is some remaining concern about the inability to quantify H3K9me2 levels by Western Blot. While the ChIP results (Figure 5H) are consistent with the model, it would be helpful to know the abundance of H3K9me2. It sounds like these experiments have already been done but the results are variable, so perhaps this can be provided as supplementary data or discussed further.

3) Please clarify the model in Figure 5—figure supplement 2A. It appears to be incorrect for the wild type. We know now that the proper wild-type control, i.e. wild-type siblings from the original *wdr-5* cross, does not have an extended lifespan. Or clarify that they are referring to wild type animals from a thaw.

4) The new Abstract is repetitive and would benefit from additional editing.

---

## [Author Response]

Essential revisions:1) Because the effects the authors are reporting are highly dependent on experimental technique – for example the reported changes in wdr-5 lifespan occur over many generations in the lab – it is critically important that these experiments are well-controlled. The authors have generally done a good job of replicating their findings multiple times. However, it is unclear why they used N2 from frozen stock as their control when the wdr-5 mutants came from a heterozygote cross. A more appropriate control would have been the wild-type siblings from the original wdr-5 cross (outcrossed for more than 5 generations to eliminate the inherited longevity effects as in Figure 1). Those generation-matched wild-type siblings could have been compared to wdr-5 mutants. Completing this experiment, at least for an early generation time-point, would validate the authors' observations that wdr-5 mutants are not long-lived until later generations compared to generation-matched wild-types while avoiding the problem of recently-thawed worms which themselves have changes in longevity over time.

We have now performed the experiments suggested by the reviewers. Our previous data indicated that something about the freeze-thaw process initially increases the longevity of wild-type worms. By middle generations, wild-type animals reach a steady state with slightly shorter lifespan. In contrast, wild-type animals coming from an outcross of *wdr-5* or *jhdm-1* mutants have not gone through a freeze-thaw. As a result, we would not expect them to have an increased lifespan. Consistent with this expectation, we find that wild-type animals coming from an outcross of *wdr-5* mutants are indistinguishable from their corresponding generation-matched mutant population (added to manuscript as Figure 2—figure supplement 1). This result validates our observations that *wdr-5* mutantsare not initially long lived, even when compared to generation-matched controls from the outcross. In addition, we compared these populations to early and late generation wild-type populations from a thaw in the same experiment. As expected, wild-type animals from the *wdr-5* outcross have a shorter lifespan than early-gen wild-type populations from a thaw, and their short lifespan resembles that of late-gen wild-type animals from a thaw.

We also made the same comparison with *jhdm-1* mutants. Similar to what we found with *wdr-5* mutants, the lifespan of early-gen wild-type populations from a *jhdm-1* outcross are indistinguishable from late-gen wild-type animals from a thaw (added to manuscript as Figure 4—figure supplement 1). This validates our findings that *jhdm-1* mutants do not reach maximum longevity until after a few generations. Consistent with our previous observations, *jhdm-1* mutants at this early generation (F2) have a slight increase in lifespan, similar to early generation wild-type populations from a thaw.

2) The author's model that wdr-5 mutation only confers a longevity benefit after many generations or when compared to late-generation wild-type is not sufficient to explain the previous observation that wdr-5 RNAi leads to increased longevity in worms within one generation (Greer et al., 2010). The authors should attempt to reproduce this finding, perhaps in early and late-generation wild-type worms, and comment on how their model can incorporate the RNAi phenotype of wdr-5. It is necessary to provide evidence that explains why the RNAi, in only one generation, is enough to extend lifespan meanwhile the wdr-5 mutants, after outcross, do not have this phenotype in early-gen populations. The type of food (in the paper, animals were grown on OP50 bacteria and for the RNAi is on HT115 bacteria) or that wdr-5 is in an operon (CEOP3136) could explain this difference.

In the original paper (Greer et al., 2010), the Brunet lab reported that a single generation of exposure to *wdr-5* RNAi increases lifespan. We agree with the reviewers that the RNAi result is not consistent with our observation that it takes many generations for *wdr-5* mutants to obtain longevity. To resolve this contradiction, we have now assessed lifespan in animals undergoing *wdr-5* RNAi. Consistent with our transgenerational *wdr-5* lifespan results, we find maintaining animals on *wdr-5* RNAi for a single generation does not confer longevity (see Author response image 1). It is unclear why the Brunet lab originally obtained different results. However, after following these populations for twelve consecutive generations on *wdr-5* RNAi, we do observe a slight increase in longevity (*P* = 0.039, shown in Author response image 1). This demonstrates that *wdr-5* RNAi may be able to affect lifespan, and lends further support to our model that lifespan is only affected after multiple generations without WDR-5 activity.

3) Related to #1, the wild-type lifespans throughout the paper are extremely variable which make the data difficult to interpret. Although C. elegans wild-type lifespan is known to be variable, the typical range at 20°C is between 17-20 days, and the authors present several figures with average wild-type lifespans of 23-24 days (Figure 1—figure supplement 1), whereas other figures have much shorter average lifespans. The differences do not seem to be only due to the change in generation after a thaw, as in Figure 1 the mid-generation lifespans for wild-type are 23-24 days, at which time the authors showed in Figure 2 that wild-type lifespan should already have decreased. Given that the authors expect mid-generation worms to no-longer have this extended lifespan after recovering from a thaw, they should attempt to explain these abnormally high results.

During one of the many times that we replicated *wdr-5* mutant’s transgenerational increase in lifespan, the wild-type and *wdr-5* mutants were longer lived than normal. This was originally included as Figure 1. It is unclear why this was the case, but we did not observe this in other repeats of the experiment. To avoid any confusion, Figure 1 has now been replaced with another replicate where wild-type median lifespan resembles the typical range. We thank the reviewers for pointing this out.

4) In the authors' working model (Figure 5—figure supplement 2), they propose that H3K9me2 levels are decreasing over generations in wild-type animals but become stable or elevated in wdr-5 mutants and their genetically wild-type F1-F4 descendants (Figure 5—figure supplement 2B). However, ChIP-seq experiments without a spike-in control do not allow direct quantitative comparison of H3K9me2 levels between samples. The metaplots of H3K9me2 ChIP-seq signal in Figure 3 and Figure 4 only indicate redistribution of H3K9me2 within a given sample but not changes in absolute quantity. To validate their model, the authors should perform ChIP-seq experiments with a spike-in control. Alternatively, they could also do Western blotting to measure the global levels of H3K9me2 in early/mid/late-generations of wild-type, wdr-5, and jhdm-1 mutants and genetically wild-type descendants of wdr-5 (F1/F3/F5 in Figure 5B—figure supplement 2).

We agree with the reviewers that, without a spike-in control, we cannot quantify an absolute increase in H3K9me2. However, we have performed ChIP-seq experiments across generations in multiple mutants, which has allowed us to clearly demonstrate a relative increase in H3K9me2 in long-lived mutants. We developed our model based on the relative increase in H3K9me2 enrichment, rather than absolute levels. Importantly, because this phenomenon occurs across large parts of the genome over many generations, we do not expect that every transgenerational population experiences precisely the same genomic changes. As a result, precisely quantifying an absolute increase at individual loci is unlikely to be meaningful. To more accurately reflect the relative increase, we have included the longer form of our discussion (formerly included in the supplement), and amended the text.

Following the reviewers’ suggestion, we tried to quantify global changes in H3K9me2 across generational time using western blots. However, by western blot, global H3K9me2 levels are very low, and we only can detect robust bands when the histone extraction is completely optimal. When we do reliably detect bands, we consistently observe increased H3K9me2 in *wdr-5* and *jhdm-1* mutants, as we reported. Therefore, we are confident in reporting that these mutants have a global H3K9me2 increase. However, by western blot the level of increase varies both between samples and between experiments. As a result, we are not comfortable using this technique to quantify a gradual increase across generations.

5) The authors attempted to claim that H3K9me2 may be the inherited mark that accounts for the inheritance of longevity in wild-type offspring of COMPASS complex mutants. They said that H3K4 is unlikely to be the inherited mark, which is presumably based on evidence from the Greer et al., 2011 paper that shows that wild-type descendants of wdr-5 mutants regain their H3K4me3 global levels even though they are still long-lived. However, if the authors want to claim that H3K9me2 may be this inherited mark, they need to show that unlike H3K4me3, H3K9me2 levels do not return to normal in wild-type offspring of wdr-5 and jhdm-1 mutants. This is a critical experiment and would make their paper much more convincing. Specifically, the authors should immunoblot for H3K9me2 (and H3K4me3 to confirm previously published work) in wdr-5 and jhdm-1 mutants, wild-type long-lived offspring, and normal wild-type worms as suggested in the previous comment.

We agree with the reviewers that if H3K9me2 is indeed the factor enabling longevity in descendants of long-lived *wdr-5* and *jhdm-1* mutants, we would expect the following: high levels of H3K9me2 would be retained in long-lived wild-type descendants, and this increase would revert to normal levels after five generations, co-incident with reversion of lifespan.

To examine this possibility we examined H3K9me2 by ChIP-PCR in wild-type descendants of long-lived *jhdm-1* mutants. We chose to use *jhdm-1* mutants because these mutants have more H3K9me2 enrichment than *wdr-5* mutants, which provides a wider dynamic range in which to observe any potential changes. We chose six H3K9me2 peaks from our ChIP-seq data that had high levels of H3K9me2 in long-lived *jhdm-1* populations (added to the manuscript as Figure 5H). Consistent with our ChIP-seq data, we find that all six loci had elevated H3K9me2 levels in long-lived *jhdm-1* mutants compared to wild-type. At three loci, we find that H3K9me2 is also high in long-lived F4 wild-type descendants of long-lived *jhdm-1* mutants, whereas levels reverted to normal in F7 wild-type descendants of *jhdm-1*. This finding provides molecular evidence that H3K9me2 is epigenetically inherited in long-lived populations. At another locus, H3K9me2 was enriched in F4 WT (*jhdm-1*) in one of two replicates. In two other loci, H3K9me2 levels are not inherited. Altogether, these data are consistent with H3K9me2 being the inherited factor that maintains longevity in wild-type descendants of *jhdm-1* mutants. Furthermore, these data suggest that the inheritance of longevity may be mediated only by a subset of loci with elevated H3K9me2 levels in *jhdm-1* mutants, rather than by a global increase in H3K9me2.

6) It seems like the authors try to make a correlation between levels of H3K9me2 enrichment and lifespan phenotype. However, this explanation does not always fit the data. For example, wild-type, wdr-5, and jhdm-2 animals all have decreased H3K9me2 enrichment at ChIP-seq peaks and from early to late generations. However, wdr-5 and jhdm-2 become long-lived at late-generations while wild-type worms have shortened lifespan. The authors should attempt to reconcile these patterns more clearly in their Discussion and avoid presenting the data as clear-cut. Additionally, it could be helpful to do a more gene-specific analysis, as perhaps the authors are missing the most important regions by looking at the average plots of all peaks, or even all germline genes. For example, the authors could look at differential peaks between wild-type and wdr-5 mutants, and check for functional categories of the nearest genes to break down this analysis further in an informative way.

We agree with the reviewers that since we do not know which loci are responsible for longevity, it is not possible to say that H3K9me2 levels absolutely correlate with lifespan in our populations over generational time.

Between early- to mid-gen *jhdm-1* mutant populations, we observe a true increase in H3K9me2 enrichment, which correlates well with the lifespan increase that occurs during the same time period. However, as the reviewers correctly identified, in *wdr-5* mutants H3K9me2 enrichment changes over generational time in ways that are not always reflected by lifespan. We have added additional text to the Discussion to more accurately reflect this nuance. For example, H3K9me2 enrichment decreases further from mid- to late-gen wild-type populations, as lifespan remains unchanged. Likewise, in *wdr-5* mutants, H3K9me2 enrichment decreases from early- to mid-gen populations as lifespan remains unchanged. Nevertheless, whenever lifespan changes significantly, we observe a corresponding change in H3K9me2 enrichment. For example, lifespan decreases in wild-type between early- and mid-gen populations, which correlates with an H3K9me2 decrease. Similarly, the lifespan increase in *wdr-5* mutants occurs between mid- and late-gen populations, which is when we observe a large increase in H3K9me2 enrichment. Therefore, among all transgenerational populations examined, whenever we observe a significant change in lifespan, we also find a corresponding change in H3K9me2 enrichment. This general trend is consistent with a model where elevated H3K9me2, perhaps only at a subset of loci, mediates longevity.

7) Figure 4D is difficult to interpret because all wild-type worms used were late-generation according to the supplementary table. If the authors want to claim there is a gain in jhdm-1 lifespan similarly to wdr-5 lifespan over generations, they should repeat this experiment with the appropriate generation-matched controls. Alternatively, they should make it very clear in the text that they used late-generation wild-type animals, as it currently reads as if there were generation-matched controls.

In Figure 2, we demonstrated that wild-type animals initially have longer lifespans after a thaw, with lifespan decreasing over a few generations until it reaches a steady state. To avoid this complication in our analysis of *jhdm-1* mutants, we chose to compare them to late-gen wild-type populations that had already researched a steady state lifespan. We have amended the text to clarify our reasoning. Importantly, in Figure 4D, *jhdm-1* mutants are initially slightly longer lived than steady state late generation wild-type populations. If we had used generation-matched wild-type populations (as was suggested in Comment 1), the lifespan of *jhdm-1* mutants would have resembled that of wild-type populations (added to the manuscript as Figure 4—figure supplement 1). Therefore, using generation-matched populations would have only lent further support to our interpretation that *jhdm-1* mutants acquire longevity over generational time (rather than contradicted our interpretation).

8) Figure 5E-G is also difficult to interpret because the authors are missing several important controls. The WT(WT) used does not appear to be a descendent from a control cross as was done for Figure 5B and C, so it is unclear whether the WT(WT) lifespan is really a proper control. Additionally, for Figure 5E there is no met-2 mutant control for lifespan. Since the authors claim that met-2 mutants are usually short-lived, there should be a met-2 mutant control to test whether being descended from a wdr-5 mutant could allow for increased lifespan relative to the shortened met-2 lifespan, not just wild-type lifespan. Additionally, the only replicate for Figure 5F shows no statistically significant difference between WT(jhdm-1) lifespan and WT lifespan, making the authors' claims in Figure 5F somewhat weak. Adding an additional replicate would strengthen this claim. Importantly, there appears to be no replicate of Figure 5G. In order to keep Figure 5F and G, an additional replicate needs to be added to Figure 5G at the minimum.

We apologize for the confusion regarding the WT (WT) control in Figure 5E. It was indeed derived from a control cross (shown on the right part of Figure 5A), as the reviewers suggested it should be. We originally left out the cross from the schema in Figure 5D for simplicity. We have now amended Figure 5D to add the ancestry of this control population.

During a replicate of Figure 5E, we did perform a *met-2 (met-2)* control to compare the F3 *met-2 (wdr-5)* populations in Figure 5E, as well as WT (*wdr-5)* and *wdr-5 (wdr-5)* controls. We originally omitted these curves for simplicity, but have added the comparison to the manuscript as Figure 5—figure supplement 1. The lifespan of the *met-2 (met-2)* control was shorter than wild-type and resembled that of *met-2 (“wdr-5”).*

We thank the reviewers for the suggestions regarded Figures 5F and 5G. We have added two additional replicates of each to Supplementary file 3.

9) jhdm-1 is predicted to specifically demethylate H3K36 instead of H3K9me2. While it is interesting that H3K9me2 enrichment patterns change in jhdm-1, it could be an indirect consequence of the alterations of H3K36 methylation. Moreover, jmjd-1.2 is the putative H3K9/H3K27 demethylase. The authors should include the jmjd-1.2 mutants in the genetic analysis to rule out that jmjd-1.2 is the more relevant demethylase.

We investigated the lifespan of *jhdm-1* mutants because JHDM-1 is a ortholog of *S. pombe* Epe1, which affects H3K9me levels and is required for the transgenerational inheritance of this mark. Consistent with their similarity, both our western blot data and ChIP-seq data suggest that H3K9me2 levels increase in *jhdm-1* mutants. During our original experiments, we also sought to examine lifespan in *jmjd-1.2 (ok3628)*, but we were unable to complete the experiments because homozygotes are embryonic lethal.

10) The model that the WDR-5/MET-2/JHDM-1 pathway induces transgenerational inheritance of longevity through germline could be strengthened. Western blot of H3K9me2 in dissected gonads of different backgrounds/conditions and a germline rescue for met-2 in the wdr-5 background across generations will help this.

There are two reasons that we concluded that the transgenerational effects on H3K9me2 levels primarily occur through the germline. In their initial experiments, the Brunet lab demonstrated that longevity in COMPASS mutants requires a proliferating germline. Additionally, for a phenotype to be inherited transgenerationally, it must be mediated through the germline. However, to more accurately reflect the work directly presented in this manuscript, we have removed “germline” from the title. We have also included the longer version of our discussion (originally included in the supplement), in which we make the connection between the germline and transgenerational inheritance more explicitly.

[Editors' note: further revisions were requested prior to acceptance, as described below.]The manuscript has been improved but there are some remaining issues that need to be addressed before acceptance, as outlined below:1) The wdr-5.1 RNAi data should be included as supplementary data. An obvious question that the readers would have after reading about the wdr-5 mutant data would be how they could be reconciled with the previously published RNAi data. The authors should include their own wdr-5 RNAi data, to indicate that at least under their experimental condition, the RNAi data are not inconsistent with their mutant data.

We have now included the RNAi data as Figure 1—figure supplement 3, amended the text to discuss the figure, and added the data to Supplementary file 3.

2) There is some remaining concern about the inability to quantify H3K9me2 levels by Western Blot. While the ChIP results (Figure 5H) are consistent with the model, it would be helpful to know the abundance of H3K9me2. It sounds like these experiments have already been done but the results are variable, so perhaps this can be provided as supplementary data or discussed further.

Because global H3K9me2 levels are very low, by western blot, the level of increase varies both between samples and between transgenerational experiments. However, when we do reliably detect bands, we consistently observe increased H3K9me2 in *wdr-5* and *jhdm-1* mutants, as we reported. Therefore, we are confident in reporting that these mutants have a global H3K9me2 increase.

We now mention the difficulty of quantifying minute changes across generational time by western in our Results. Because we are not comfortable using this technique to quantify a gradual increase across generations, we are not including the westerns as supplementary data. Instead, we feel that the more quantitative ChIP-qPCR results presented in Figure 5H demonstrate the heritability of H3K9me2 across generations.

3) Please clarify the model in Figure 5—figure supplement 2A. It appears to be incorrect for the wild type. We know now that the proper wild-type control, i.e. wild-type siblings from the original wdr-5 cross, does not have an extended lifespan. Or clarify that they are referring to wild type animals from a thaw.

We have amended the figure legend to clarify that Figure 5—figure supplement2A represents wild-type animals recovered from a thaw.

4) The new Abstract is repetitive and would benefit from additional editing.

We thank the reviewers for this suggestion – we have edited the Abstract as suggested.